# Quantitative modeling identifies critical cell mechanics driving bile duct lumen formation

**Paul Van Liedekerke**[1,2,4☻‡*], **Lila Gannoun**[2☻], **Axelle Loriot**[2], **Tim Johann**[3], **Frédéric P. Lemaigre**[2‡], **Dirk Drasdo**[1,3,4‡*]

**1** Inria Saclay Île-De-France, Palaiseau, France, **2** de Duve Institute, Université Catholique de Louvain, Brussels, Belgium, **3** Leibniz Research Centre for Working Environment and Human Factors at the Technical University Dortmund, Dortmund, Germany, **4** Inria de Paris & Sorbonne Université LJLL, Paris, France

☻ These authors contributed equally to this work.
‡ These authors are joint co-senior authorship on this work.
\* Paul.Van_Liedekerke@inria.fr (PVL); Dirk.Drasdo@inria.fr (DD)

**Data Availability Statement:** Immunostainings: this data is entirely contained within the manuscript. The RNA-seq data have been deposited in the Gene Expression Omnibus (GEO)

## Abstract

Biliary ducts collect bile from liver lobules, the smallest functional and anatomical units of liver, and carry it to the gallbladder. Disruptions in this process caused by defective embryonic development, or through ductal reaction in liver disease have a major impact on life quality and survival of patients. A deep understanding of the processes underlying bile duct lumen formation is crucial to identify intervention points to avoid or treat the appearance of defective bile ducts. Several hypotheses have been proposed to characterize the biophysical mechanisms driving initial bile duct lumen formation during embryogenesis. Here, guided by the quantification of morphological features and expression of genes in bile ducts from embryonic mouse liver, we sharpened these hypotheses and collected data to develop a high resolution individual cell-based computational model that enables to test alternative hypotheses in silico. This model permits realistic simulations of tissue and cell mechanics at sub-cellular scale. Our simulations suggest that successful bile duct lumen formation requires a simultaneous contribution of directed cell division of cholangiocytes, local osmotic effects generated by salt excretion in the lumen, and temporally-controlled differentiation of hepatoblasts to cholangiocytes, with apical constriction of cholangiocytes only moderately affecting luminal size.

## Author summary

The initial step in bile duct development is the formation of a biliary lumen, a process which involves several cellular mechanisms, such as cell division and polarization, and secretion of fluid. However, how these mechanisms are orchestrated in time and space is difficult to understand. Here, we built a computational model of biliary lumen formation which represents every cell and its function in detail. With the model we can simulate the effect of biophysical aspects that affect duct formation. We have tested the individual and combined effects of directed cell division, apical constriction, and osmotic effects on lumen expansion by varying the parameters that control their relative strength. Our

database and assigned the identifier GSE163062. In Fig 3C, all relevant data are in the graph. All the simulation data created in Figs 7, 8, 9 and 10 are simulation data can be plotted using the accompaying python script (delivered as supplementary information). This paper entered editorial review prior to the code availability policy of 30/03/2021.

**Funding:** The work of PVL and DD were supported by LiSyM (nr 031L0045) by BMBF (http://www.bmbf.de). PVL, DD and FPL acknowledge iLITE (nr ANR-16-RHUS-0005-16) by ANR (http://www.anr.fr). The work of FPL was supported by D.G. Higher Education and Scientific Research of the French Community of Belgium (grant ARC 15/20-065, http://www.recherchescientifique.be/). FPL and AL acknowledge the Fonds de la Recherche Scientifique FRS-FNRS (Belgium; grants T.0158.20 and J.0115.20, https://www.frs-fnrs.be/). L.G. holds a PhD fellowship from the Fonds pour la Formation à la Recherche dans l'Industrie et dans l'Agronomie (Belgium: grant 1.E071.18, https://www.frs-fnrs.be/). The funders had no role in study design, data collection and analysis, decision to publish, or preparation of the manuscript.

**Competing interests:** The authors have declared that no competing interests exist.

simulations suggest that successful bile duct lumen formation requires the simultaneous contribution of directed cell division of cholangiocytes, local osmotic effects generated by salt excretion in the lumen, and temporally-controlled differentiation of hepatoblasts to cholangiocytes, with apical constriction of cholangiocytes only moderately affecting luminal size.

## Introduction

The liver is composed of multiple repetitive anatomical and physiological units, the liver lobules. A liver lobule is approximately hexagonal in shape and is fed by oxygen rich blood from the hepatic artery, and oxygen-poor blood from the portal vein. Inside the lobule the blood flows through complex network of capillaries with a design that promotes a maximal exchange of molecules between blood and hepatocytes until it drains into the central vein. Bile is eliminated via the bile canaliculi, and collected by biliary ducts that carry the bile to the gallbladder and eventually to the intestine. Defects in the biliary duct system resulting from defective development or from disease, e.g. by ductal reaction [1] can significantly impact life quality and survival of a patient. A deep understanding of the processes underlying lumen formation during bile duct development is crucial to intervene in such defective cases, and can furthermore be used to engineer functional liver tissue in vitro.

In embryonic liver, cholangiocytes and hepatocytes differentiate from bipotent hepatoblasts. Differentiation of cholangiocytes is spatially restricted around the mesenchyme associated with the branches of the portal vein [2, 3]. The conversion of hepatoblasts to cholangiocytes is induced by cell-cell contacts between the mesenchymal cells and adjacent hepatoblasts, as well as by factors secreted by the mesenchyme which target receptors at the membrane of the hepatoblasts [4].

Hepatoblasts that have differentiated to cholangiocytes initially constitute a "ductal plate", which is a discontinuous and single-layered sheet of cholangiocytes located near the portal mesenchyme (Fig 1A and 1C). Duct morphogenesis is then initiated by the appearance of several lumina delineated on the portal side by ductal plate cholangiocytes, and on the parenchymal side by hepatoblasts. Therefore, the initial ductal structures are lined by two distinct cell types. When duct formation proceeds, the hepatoblasts which delineate the parenchymal side of the lumina progressively differentiate to cholangiocytes [3–8].

Bile duct lumen formation proceeds according to a cord hollowing process which, consists in the appearance of a lumen between adhering cells [5, 8, 9]. Lumenogenesis implies that the cells which delineate an apical lumen coordinately undergo apico-basal polarization and develop adhesive junctions [10, 11]. A space can form between the apical surfaces of cells that face each other, as a result of repulsive forces, exocytosis of vesicles containing luminal components, or accumulation of fluid. Fluid leakage should then be prevented by junctional complexes between the cells that delineate the apical luminal space. Putting these notions in the context of biliary lumen formation, earlier work has shown that cholangiocytes develop apical poles when they start forming the ductal plate (Fig 1C). Polarization is initiated in single cholangiocytes and extends to adjacent cholangiocytes, likely via Notch and Neurofibromin 2 signaling. The cholangiocytes then coordinately and collectively contribute to delineate a lumen together with hepatoblasts [5, 8, 12]. Still, the driving forces allowing lumen formation remain unclear.

Computational models offer means to simulate a set of hypothesized mechanisms free from unknown influences and identify whether this set is either able or fails to explain experimental

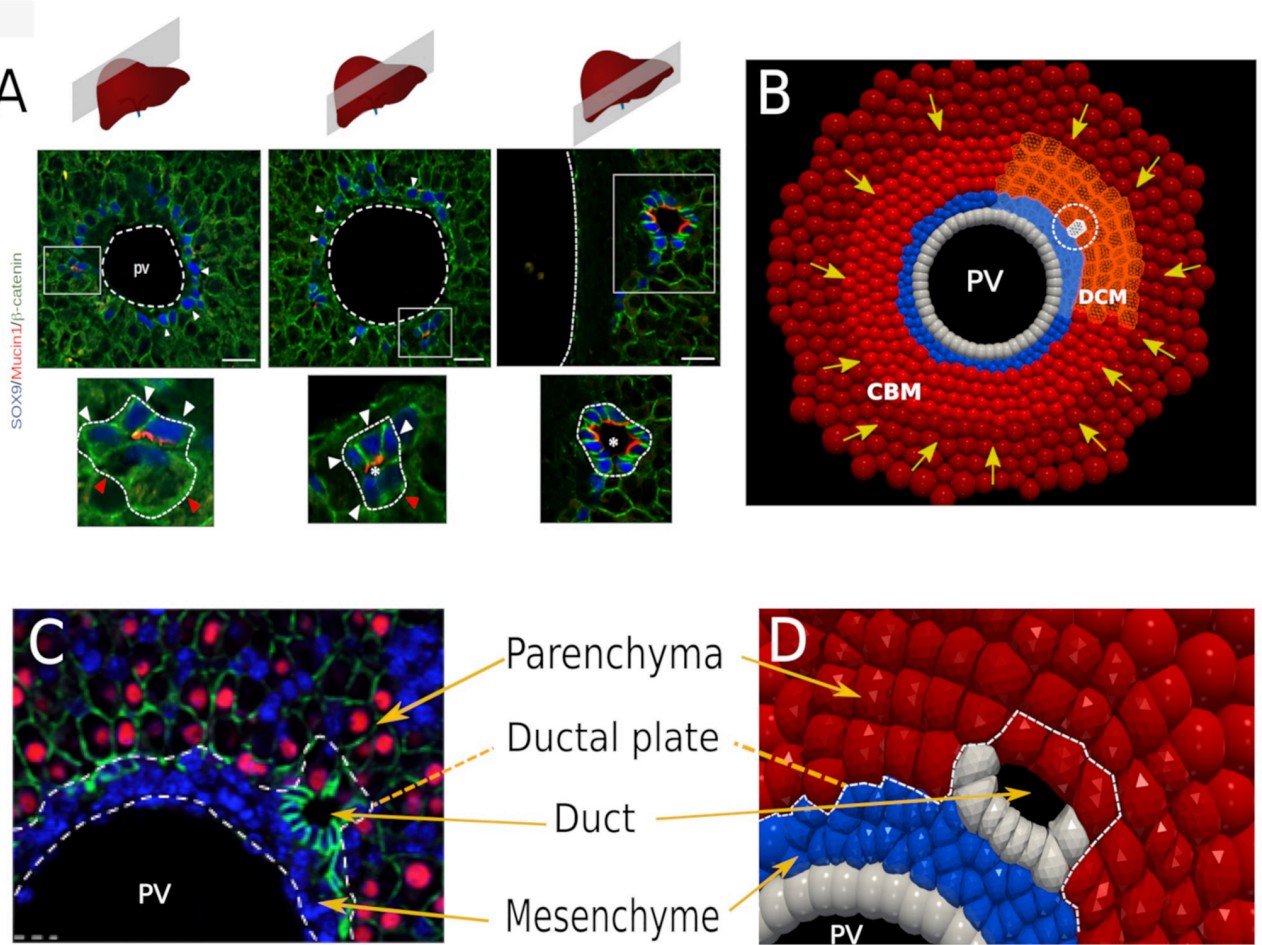

**Fig 1. Experimental images and computational approach of bile duct lumen formation.** (A) The bile ducts mature from the hilum (right panel) to the periphery of the liver lobes (left panel), as illustrated with sections made at several distances from the hilum in mouse embryonic day (E) 18.5 liver. Nascent lumina are identified with Mucin-1 staining. White arrowheads point to SOX9-expressing cholangiocytes; red arrowheads point to hepatoblasts delineating nascent lumina. Size bar is 20 $\mu m$. (B) Snapshot of the equivalent in silico model showing the CBM and DCM regions. The initial cholangiocyte is marked by a dashed circle. The yellow arrows indicate the background pressure forces. (C) Tissular organization of the developing liver. The portal vein (PV) is delineated by an endothelium and is surrounded by mesenchyme. Cholangiocytes form a discontinuous layer of cells called ductal plate. When lumina start to form they are delineated by cholangiocytes and Hepatocyte Nuclear Factor 4 (HNF4)-expressing hepatoblasts (red). Epithelial cells are identified by beta-catenin staining (green); DAPI stains nuclei (blue). The left panel illustrates a section through a mouse liver at E18.5; the right panel (D) shows the tissular architecture in a close-up of the in silico model.

observations, which in the latter case indicates missing or false hypotheses. Over the past years, a class of computational models, called agent-based models (ABM), have been increasingly deployed to better understand the role of mechanics in tissue growth, regeneration and development. In ABMs of tissues, each cell is represented individually and is characterized by its behavior as well as by state changes. For example, a cell may be able to grow, divide, migrate, and undergo state changes, whereby the state might represent a cell type, activation state, etc. Cell dynamics and state are formalized by mathematical rules or equations. Parameters at the cell-level may be functions of intracellular molecular processes such as e.g. signaling events. ABMs have been explored in discrete and continuum space. In continuum space, a first large class of ABM are called center-based models (CBM), in which a cell shape is rigid and geometrically approximated by an object that represents a coarse-grained description of cell shape

only in a statistical sense by occupying a region in space where most of the cell volume is located [13–23]. A second class can be largely categorized as "deformable" models, e.g [24–38], in which cell shape is resolved explicitly, hence permitting a much more accurate representation of the interplay of cell shape and forces on the cell. The use of deformable models may be considered as mandatory in cases where cell shape is assumed to be a critical determinant, for example, driving or controlling local cell arrangements.

At the onset of biliary morphogenesis, very small lumina and a high variability in cell shapes are observed close to the formed lumina, meeting precisely the conditions to use a deformable model. For these reasons, we employed a high resolution model for the local arrangement of cells forming the lumen, and a CBM far from the lumen-forming region (Fig 1B). We called this model the "Deformable Cell Model" (DCM). The cell surface is flexible and its shape adapts naturally to the environmental constraints.

The DCM allows coupling to CBMs, because in both models spatial displacements are based on the calculation of physical forces or mechanisms that can be represented by a physical force, as e.g. compression forces emerging from cell proliferation in a tissue. The coupling permits simulation of tissue organization processes in a hybrid mode, where the CBM can represent cells for which a lower resolution is sufficient, while the DCM accounts for the regions where high resolution is required [37] (see S1 Text for more details).

We extended the recently established model [37], that can simulate growth and proliferation in monolayers, spheroids, and regeneration after drug-induced damage in a liver lobule, with the ability of a cell to acquire a polarity defining the polar direction, and distinguish an apical and basal side. Each cell may be subject to active cortical tension and apical constriction, and can form Tight junctions (TJ) with other cells. Finally, we allowed the inclusion of local osmotic effects that potentially arise in the lumen area (see Computational models).

In this work, we have included three different mechanisms that are hypothesized to contribute to lumen formation. We first investigated the individual effects of each of these mechanisms, namely: coordinated cell division, apical constriction and osmotic effects.

Our simulations showed that each of these mechanisms can create a lumen in an idealized system without boundary conditions. In a second stage, guided by the quantification of morphological features and expression of genes in developing bile ducts of embryonic mouse liver, we constructed an in silico system representing a portion of the lobule containing the portal vein and surrounding tissue (see Fig 1B and 1D).

Using this architecture we have simulated the effects of the aforementioned mechanisms both individually and combined. We found that contrary to the idealized system, a coordination of these mechanisms is necessary to initiate lumen formation and allow further lumen growth.

## Materials and methods

### Ethics statement

Mice received humane care and the research protocol was approved by the Animal Welfare Committee of the Universite Catholique de Louvain with number 2018/UCL/MD/014.

### Animals

All mice (CD1 strain) were housed under a 12h light/12h dark cycle in individually ventilated cages supplied with RM3 chow (#801700, Tecnilab, Someren, Netherlands), acidified water and polyvinyl chloride play tunnels. Sox9-GFP mice have been described [39].

## Cell isolation and RNA extraction

Pools of 15 livers from Sox9-GFP mice at E16.5 or E18.5 were dissected, minced and dissociated in DMEM-F12 (#31870–025, Gibco, Life technologies, Lederberg, Belgium) containing 1 mg/ml collagenase IV (#43E14253, Worthington, New Jersey, USA), 1 mg/ml dispase (#17105–041, Gibco, Life technologies) and 0.1 mg/ml of DNAse I (#11284932001, Roche, Mannheim, Germany) for 30 min at 37˚C. Digestion was stopped by adding an equal volume of 10% foetal bovine serum in phosphate buffered saline (PBS) and cells were resuspended in 2 mM EDTA, 0.5% bovine serum albumin in PBS and filtered on 40 μm cell strainer (#087711, Fisher Scientific, Merelbeke, Belgium). CD5+/CD45R+/CD11b+/7/4+/Ly-6G/C+/Ter119+ hematopoietic cells were eliminated by MACS separation using the "Lineage Cell depletion Kit" (#130090858, Miltenyi Biotech, Paris, France). Sox9-GFP+ cells were isolated by fluorescence-activated cell sorting (BD FACSAria III, sample and collection tubes maintained at 4˚C). Total RNA was isolated using RNAqueous Micro kit (#AM1931, Invitrogen, Carlsbad, CA, USA) according to manufacturer's protocol. RNA quality was evaluated using the Agilent RNA 6000 Pico Kit (Agilent Technologies, Santa Claa, CA, USA) and Bioanalyzer (Agilent Technologies) for measuring concentration and calculation of RNA integrity number (RIN).

## RNA sequencing and bioinformatics workflow

Read quality control was performed using FastQC software v0.11.7 (Babraham Institute, Cambridge, UK). Low quality reads were trimmed and adapters were removed using Trimmomatic software v0.35 (RWTH Aachen University, Aachen, Germany). Reads were aligned using HISAT2 software v2.1.0 (Johns Hopkins University School of Medicine, Center for Computational Biology, Baltimore, MD, USA) on GRCm38 mouse genome. Gene counts were generated using HTSeq-count (v0.5.4p3) software and gencode.vM15.annotation.gtf annotation file and raw counts were further converted in transcripts per million (TPM).

## Immunofluorescence and imaging

Immunofluorescence stainings were performed on 6 $\mu$m sections of formalin-fixed paraffin-embedded tissues. Tissue sections were deparaffinized 3x 3 min in xylene, 3 min in 99%, 95%, 70% and 30% ethanol and deionized $H_2O$. Antigen unmasking was performed by micro-wave heating for 10 min in 10 mM sodium citrate pH 6. Sections were permeabilized for 5 min in 0.3% Triton X-100 PBS solution before blocking for 45 min in 0.3% milk, 10% bovine serum albumin, 0.3% Triton X-100 in PBS. Primary and secondary antibodies were diluted in blocking solution and incubated respectively at 4˚C overnight and 37˚C for 1 h. Pictures for immunofluorescence were taken with Cell Observer Spinning Disk (Carl Zeiss, Oberkochen, Germany) and Zen blue software (Carl Zeiss). Primary and secondary antibodies are described in Table 1.

## Computational models

**Deformable cell model (DCM).**   In the DCM used throughout this work the 3D cell surface was discretized by a set of nodes which are connected by viscoelastic elements [27, 35, 37]. The nodes and their connecting elements generate a triangulation of the cell surface. The total force on each node sums up all forces on that node including mechanical forces within the cortex, external cell-cell contact interaction forces, nodal forces originating from volume conservation, as well as cell migration and osmotic effects. We note here that, although the model components are essentially 3D, all simulations are performed in a 2D slice of the bile ducts by constraining movement to the $(x, y)$-plane, which may be referred to as "2.5D".

**Table 1. Antibodies used for immunostaining.**

| PRIMARY ANTIBODY | SPECIES | SOURCE | REFERENCE NUMBER | DILUTION |
|---|---|---|---|---|
| ZO-1 | Rabbit | Invitrogen | 61–7300 | 1/100 |
| SOX9 | Rabbit | Merck Millipore | AB5535 | 1/250 |
| HNF4a | Mouse (IgG2a) | R&D Systems | PP-H1415–00 | 1/350 |
| Ki67 | Mouse (IgG1) | BD Biosciences | 556003 | 1/250 |
| Mucin-1 | Armenian Hamster | Invitrogen | MA5–11202 | 1/350 |
| $\beta$-catenin | Mouse (IgG1) | BD Biosciences | 610153 | 1/1000 |
| E-Cadherin | Mouse (IgG2a) | BD Biosciences | 610181 | 1/200 |
| **SECONDARY ANTIBODY** | **SPECIES** | **SOURCE** | **REFERENCE NUMBER** | **DILUTION** |
| Alexa Fluor 488 anti-rabbit | Donkey | Invitrogen | A21206 | 1/1000 |
| Alexa Fluor 594 anti-rabbit | Donkey | Invitrogen | A21207 | 1/1000 |
| Alexa Fluor 647 anti-mouse IgG2a | Goat | Invitrogen | A21241 | 1/1000 |
| Alexa Fluor 488 anti-mouse IgG1 | Goat | Invitrogen | A21121 | 1/1000 |
| Alexa Fluor 594 anti-armenian hamster | Goat | Jackson ImmunoResearch | 127–585–160 | 1/1000 |

The dynamics of the cell is determined by an equation for each node according to Newton's second law, whereby inertia effects are negligible compared to friction forces. For any node $i$ of a cell (the cell index has been dropped here for clarity) in which $\vec{v}_i$ denotes the velocity of node $i$ (see Fig 2A), the equation reads:

$$\underbrace{\Gamma_{ns,i}\vec{v}_i}_{\text{substrate friction}} + \underbrace{\sum_j \Gamma_{nn,ij}(\vec{v}_i - \vec{v}_j)}_{\text{node-node friction}}$$

$$= \underbrace{\sum_j \vec{F}_{e,ij}}_{\text{in-plane}} + \underbrace{\sum_m \vec{F}_{m,i}}_{\text{bending}} + \underbrace{\vec{F}_{vol,i}}_{\text{volume change}} + \underbrace{\vec{F}_{rep,i} + \vec{F}_{adh,i}}_{\text{contact}} + \underbrace{\vec{F}_{mig,i}}_{\text{migration}} + \underbrace{\vec{F}_{osm,i}}_{\text{osmosis}}. \qquad (1)$$

The matrix $\Gamma_{ns}$ is the cell-substrate viscous friction matrix and $\Gamma_{nn}$ the node-node friction matrix originating from viscous effects inside the cell or between two cells. The first and the 2nd term on the rhs. represent the in-plane elastic forces and bending forces in the cortex, respectively, determined by the elastic parameters of the cortex. The third term on the rhs. is a volume force controlled by the cell cytoplasm compressibility and water in/outflow. The fourth term ($\vec{F}_{adh,i}$, $\vec{F}_{rep,i}$) describes the adhesion and repulsion forces on the local surface element in presence of nearby objects such as another cell. The adhesive forces are controlled by the specific adhesion energy parameter $W$ which is determined by the cadherin density on the cell surface. The repulsive forces prevent that the surfaces of two different cells can interpenetrate. Both terms are calculated from the Maugis-Dugdale theory of adhesive bodies. We assume that cell migration can be present if cells are locally surrounded by ECM. A resulting force $\vec{F}_{mig}$, un-directed and mimicked by a Brownian motion term with zero mean and uncorrelated in time, is distributed equally on all nodes. The final term describes external pressure forces due to osmotic effects if these are present (for more details on the force terms, see S1 Text).

Cell division is mimicked by a process in which the mother cell is replaced by a pair of daughter cells which initially are both contained by the mother cell envelope (see Fig 2G), as described in S1 Text. The direction in which the cell divides can be chosen randomly (as is the

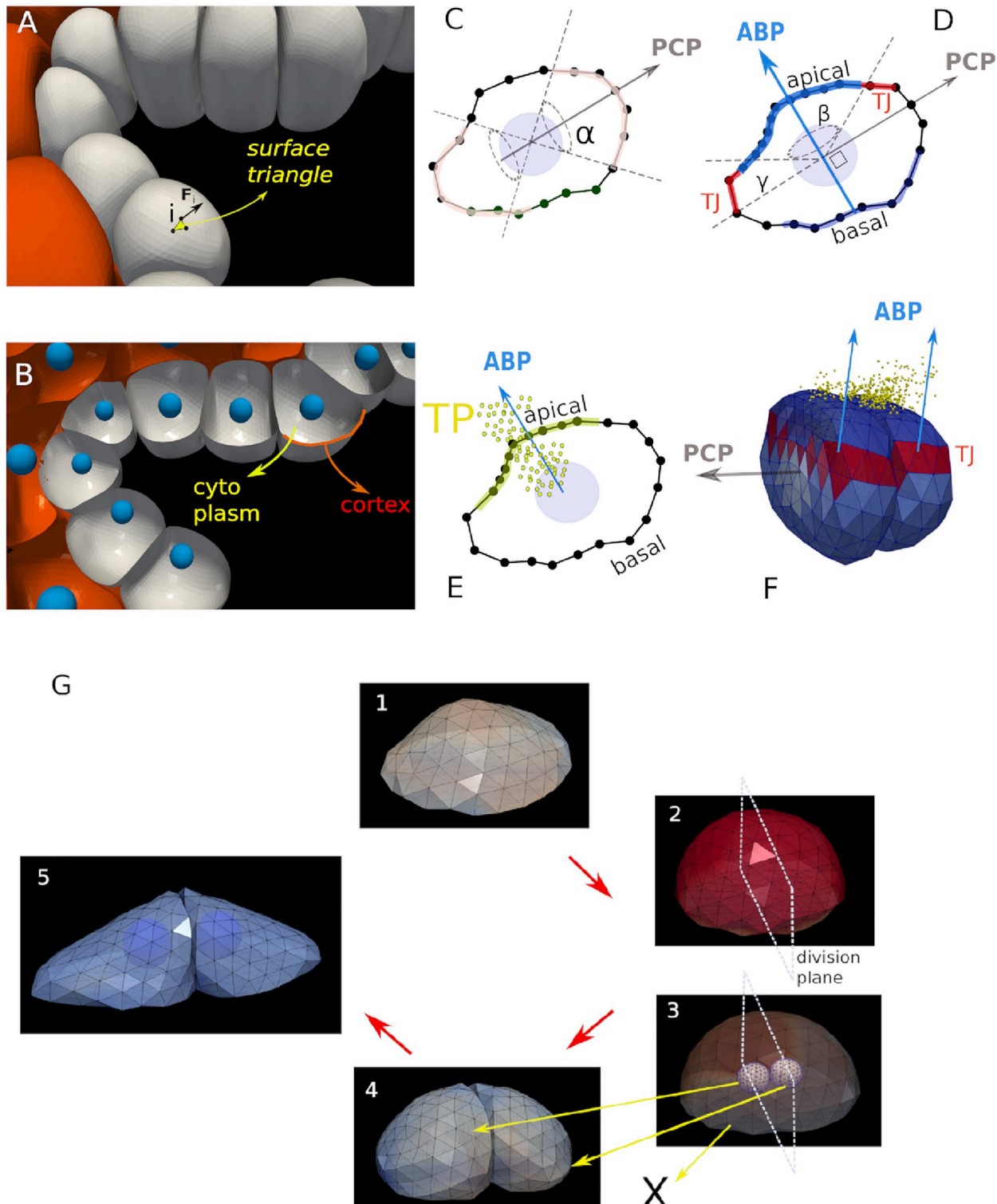

**Fig 2. The DCM functional elements.** (A) Cell surface element (triangle) with nodal force. (B) Horizontal cut section through the DCM model plane, indicating cytoplasm and cortex of the cells. C-E are cross sections through individual cells. C) 2D sketch defining the polarity vector (PCP) formed by the two cones with opening angle $\alpha$. Note that the zone of polarity is assumed to be symmetric. (D) Definition of the Apical-Basal vector (ABP). (E) Tracer Particles (TP) moving across the apical surface of the cell. (F) Tight junctions between two cells in contact represented by red colored triangles. (G) Overview of the different stages for the DCM cell division algorithm, extended from [37] 1: A cell with arbitrary shape. 2: just

before cell division, the cell rounds up. A division plane is chosen. 3: Two new smaller spherical cells are created on both sides of the plane inside the mother envelope. Both daughter cells first grow artificially fast ("sub-simulation") within the boundaries of the mother envelope, to reach their target initial volume. 4: Shortly after, the mother envelope is removed. The cells adapt to their new environment. 5: Two new growing and adhering cells have been created (nuclei are shown).

case for non polarized hepatocytes) or with a preferred direction. In the present case division perpendicular to the apico-basal axis is required to generate two daughter cells that maintain the epithelium single-layered and the division direction is along the polarization vector. As the DCM are parameterized by physical quantities, the range of its parameter values can mostly be readily estimated (see Table 2).

**New extensions to the DCM.**    Three major new extensions were made to permit model simulations for virtual tests of the key hypotheses of early bile duct formation. Firstly, a polarity vector for each cell was introduced, indicating the Planar Cell Polarity (PCP), and an apical-basal vector, indicating the Apical-Basal Polarity (ABP), see e.g. [40]. The polarity vector can determine in which direction a cell divides (as opposed to random division directions). The ABP vector determines the area of cell apical constriction (AC) as the apicial region of the cell is defined relative to the ABP vector. Secondly, the model has been extended with the capability to form apical Tight Junctions (TJ) between cells. These are limited surface zones where

**Table 2. Nominal physical parameter values for the model.** An (*) denotes parameter variability meaning that the individual cell parameters are picked from a Gaussian distribution with ±10% on their mean value. CR: Calibration Runs. Unless indicated, the cell parameters for the CBM and DCM are identical. PC: Personal communication.

| Parameter | symbol | unit | value | ref |
|---|---|---|---|---|
| **Deformable Cell Model** | | | | |
| Radius Hepatoblasts | $R_c$ | $\mu m$ | $8.8 - 12$ | observation, [45] |
| Radius Cholangiocytes | $R_c$ | $\mu m$ | $8.8 - 12$ | observation, [45] |
| Radius Mesenchyme | $R_c$ | $\mu m$ | $8.8 - 12$ | observation, [45] |
| Cycle time* | $\tau$ | $h$ | 24 | [45] |
| Cortex Young's modulus | $E_{cor}$ | $Pa$ | 1000 | [46] |
| Cortex thickness | $h_{cor}$ | $nm$ | 500 | observation |
| Cortex Poisson ratio | $v_{cor}$ | - | 0.5 | [47] |
| Cell bulk modulus | $K_V$ | $Pa$ | $750 - 2500$ | [45, 47] |
| Global cell-cell adhesion energy | $W$ | $J/m^2$ | $9 \cdot 10^{-4}$ | [45],CR |
| Apical adhesion energy | $W_{ap}$ | $J/m^2$ | $10^{-6}$ | CR |
| Nodal friction | $\gamma_{int}$ | $Ns/m^3$ | $1 \cdot 10^{-4}$ | CR |
| Cell-cell friction | $\gamma_{ext}$ | $Ns/m^3$ | $5 \cdot 10^{10}$ | [48, 49],CR |
| Cell-ECM friction | $\gamma_{ECM}$ | $Ns/m^3$ | $10^8$ | [48],CR |
| Cell-liquid friction | $\gamma_{liq}$ | $Ns/m^3$ | 500 | CR |
| Motility | $D$ | $m^2/s$ | $10^{-16}$ | [45, 50] |
| **Center Based Model** | | | | |
| Young's modulus* | $E$ | $Pa$ | 450 | [45] |
| Poisson's modulus | $v$ | – | 0.47 | [45] |
| Motility | $D$ | $m^2/s$ | $10^{-16}$ | [45, 50] |
| **Lobule** | | | | |
| Radius portal vein | $R_{pv}$ | $\mu m$ | 50 | [45] |
| Pressure lumen ducts | $P_L$ | $Pa$ | $0 - 100$ | CR |
| Pressure tissue | $P_b$ | $Pa$ | 50 | PC |

cells strongly attach to each other (see Fig 2F). Finally, to simulate osmotic effects in the extra-cellular space, the concept of "Tracer Particles" (TP) has been introduced to mimic osmotic effects caused by local differences in salt concentration. For the technical aspects of the implementation we refer to S1 Text.

The polarity vector bound to each cell can define zones of the cell surface, which are more adhesive or less adhesive (e.g. by zones with different cadherin density). Here these zones are assumed to be formed by cones centralized in the cell and aligned by the polar vector (Fig 2C). The polar surface zone on the cell is assumed to be symmetrical along both cone sides, and the angle of the cone determines the surface area (as in [13, 41]).

The ABP vector can mark a bi-conical region where later apical constriction can occur. To mimic the constriction effect which is driven by actin-myosin contractions and whereby the apical side of the cell surface contracts, the equilibrium distance between nodes are reduced to a shorter distance. As a consequence the nodes will move towards each other until a new mechanical equilibrium is reached. A distinction can be made in the model for "circumferential" constriction (ring-like zone where the cytoskeleton contracts) and "medioapical" constriction where the contraction is situated in the apical domain [42] (see S1 Text). On the opposite basal side, the cell surface remains as before, see cartoon Fig 2D. An internal structure was not needed to capture the key physics of the constriction process but could be added to the model, at the expense of more parameters (the explicit representation of the internal cytoskeleton would require the remodeling of the cytoskeleton rorganisation during apical constriction). and computational time. As the cortex is contracted on the apical side, the nodes smoothly move from the basal side to the apical side.

Tight junctions (TJ) are the areas at cell-cell contacts that have a much higher adhesion energy than the regular adhesive zones. The TJ regions were defined relative to the APB vector and by the width of the region. The TJ is represented in the model as a conical region (belt) with a certain width controlled by an angle $\gamma$ (see Fig 2D). The "belt" is part of the cell's lateral surface and separates the apical region and the basal region.

Tracer Particles (TP) are discrete particles representing clouds of molecules that can be secreted by a cell (Fig 2E). The TP can diffuse into the extracellular space without interacting with each other. They can mark a part of the surface of a cell if they come in contact with free surface area of that cell. Once a part the DCM has been marked, it can become unmarked if it comes again in contact with a part from another cell (e.g. through adhesion). If the TP represent salt ions, we use these marks to define the regions in the extracellular space where a net osmotic pressure on the cell surface is generated due to water attraction. Conceptually similar, TP could also represent signaling molecules that are sensed by nearby cells. However, since bile starts to flow around E16.5-E17.5, i.e. slightly later than the initiation of bile duct lumen formation (E15.5) [7], TP's cannot correspond to bile at the onset of lumen formation. We assume in our model that tracer particles are excreted on the apical side of the cell.

**Center based model (CBM).** The high spatial accuracy of the DCM causes longer computation times, which can limit the use of that model for large cell populations. In CBMs the individual cell shape is not represented explicitly, simplifying and approximating the results, but significantly reducing the computational time. As for the DCM, the CBM is able to mimic active migration, growth and division, and interaction with other cells or a medium. The CBM is here used to model the parts of the system that do not require knowledge of the precise cell shape. Nevertheless the CBM needs to fulfill its role as a component with a realistic behavior and, if replaced by a DCM, should not lead to different simulation results. We here use the CBM to model the boundary of the portal vein (PV), which we assume to not move during the time course of the simulation, and the hepatoblasts that do not participate in bile duct formation, but exert a certain mechanical force to the inner cells closer to the PV (Fig 1C).

As for the DCM, the movement of cells in the CBM is described by Newton's law of motion. Different from the DCM, where all forces on a large set of nodes characterizing cell shape has been considered, in the CBM only forces on the center of mass of the cell is taken into account. The center of mass position of each cell $i$ is obtained from a Langevin equation of motion, which summarizes all forces on that cell including a force term mimicking its micro-motility:

$$\Gamma_{ECM}\vec{v}_i + \sum_j \Gamma_{cc}(\vec{v}_i - \vec{v}_j) = \sum_j \vec{F}_{cc,ij} + \vec{F}_{mig,i} \tag{2}$$

The lhs. describes cell-matrix friction, cell-capsule friction and cell-cell friction, respectively. Accordingly, $\Gamma_{ECM}$ and $\Gamma_{cc}$ denote the friction tensors for cell-ECM and cell-cell friction. The first term on the rhs. of the equation of motion represents the cell-cell repulsive and adhesive forces $\vec{F}_{cc}$, which can be approximated by the Johnson-Kendall-Robert (JKR) model, approximating cells by isotropic homogeneous sticky elastic bodies that are moderately deformed if pressed against each other [43]. It is well-known that the original JKR contact model becomes inaccurate in systems with large cell densities, where it underestimates the contact forces and becomes inconsistent with the forces in the DCM [44]. Here, a modification of the JKR contact force model was calibrated with DCM using the calibration procedure. As a consequence, as long as cell shape does not largely deviate from a sphere, CBM and DCM generate statistically equivalent simulation results and can be used in a hybrid model by switching between them if necessary (see ref. [37] and S1 Text for details). The 2nd term is an active force term $\vec{F}_{mig}$, mimicking the cell micro-motility. Similar to the DCM the $\vec{F}_{mig}$ is mimicked by a Brownian motion term with zero mean value and uncorrelated in time.

## Results

### Morphological features of cells during biliary lumenogenesis

First, we present the experimental observations that support alternative hypotheses on the mechanisms controlling early lumen formation during duct development. In a second step, the three different hypotheses are tested.

At the onset of biliary lumen formation, cholangiocytes are located adjacent to the periportal mesenchyme, which separates the cholangiocytes from the endothelial cells lining the portal vein. The parenchyma is predominantly composed of beta-catenin-positive hepatoblasts, hematopoietic cells and vascular spaces (Fig 1A) [51]. Beta-catenin is here used as an epithelial marker.

Lumen formation was shown to be initiated at single cholangiocytes expressing Na+/H+ exchanger regulatory factor 1 (NHERF1) and Moesin at their apical pole [8]. Fig 1A extends this information by illustrating the expression of Mucin1 (MUC1), a cell surface glycoprotein, at the apical pole of cholangiocytes. Initially, a single cholangiocyte, identified by expression of the biliary marker SRY-related HMG box transcription factor 9 (SOX9) expresses Mucin-1 (MUC1). The expression of MUC1 then spreads to adjacent cholangiocytes, in parallel with expansion of the lumen and differentiation of hepatoblasts into cholangiocytes. The various steps of lumen formation are shown in Fig 1A at embryonic day (E) 18.5 within the same liver. This is made possible because duct formation progresses from the liver hilum to the periphery of the lobes, with more mature structures being located near the hilum and less mature structures more peripheral in the liver.

Lumen formation is associated with apico-basal polarization of the cholangiocytes. This is well illustrated by expression of NHERF1, Moesin, MUC1 and osteopontin, all at the apical pole of the cholangiocytes (refs. [5, 8] and Fig 1A). Further, when determining the presence of

tight junctions between cells delineating a developing lumen, the tight junction marker Zonula Occludens 1 (ZO1) was detectable at the junction between adjacent cholangiocytes and between a cholangiocyte and a hepatoblast in asymmetrical ductal structures (Fig 3A). In contrast, adjacent hepatoblasts delineating a developing duct lumen do not yet express detectable ZO1 near the biliary lumen, indicating that tight junction formation parallels maturation of the hepatoblasts to cholangiocytes.

**Hypothesis I: Differential proliferation rates.**   Since the portal vein is surrounded by a double layer consisting of an inner cholangiocyte and an outer hepatoblast cell layer, differential proliferation rates within the two distinct layers may create buckling forces large enough to rupture the contacts between the two layers leading to generation of a lumen. Therefore, we measured the percentage of proliferating cholangiocytes and hepatoblasts. Using SOX9/Ki67 and HNF4/Ki67 co-stainings we found that 26.0% of SOX9-positive cholangiocytes and 10.7% of HNF4-positive hepatoblasts were proliferating at E16.5 (Fig 3B) (n = 3 livers; 630 cholangiocytes and 549 hepatoblasts were analyzed). The first hypothesis studied below by the computational model is that this difference may be responsible for lumen formation.

**Hypothesis II: Apical constriction.**   Formation of a lumen in a cylindrical duct implies that the apical pole of the cells aligning the lumen is (geometrically) shorter than the opposite basal pole. We verified that this was the case. We measured the length of the apical and basal poles of cells involved in the formation of asymmetrical ductal structures (Fig 3C). Apical shortening was obvious in cholangiocytes, despite significant cell-to-cell variability. The heterogeneous morphology and circumferential expression of E-cadherin in hepatoblasts did not allow us to delineate the apical and basal sides or measure the length of the poles. However, plotting the length of the apical sides as a function of the length of the basal sides in cholangiocytes showed that all cells display apical constriction, the ratio of basal/apical length being > 1. Therefore, apical constriction whereby the apical shortening is driven by an active constriction of the apical side of the cholangiocytes, would be a candidate mechanism for the lumen formation.

**Hypothesis III: Osmosis-driven lumen formation.**   The accumulation of fluid may be a potential driving force for lumenogenesis. Such accumulation of fluid is expected to result from osmotic changes and water transport, which depend on ion transporters and water channels. To verify if ion transporters and aquaporins were expressed at the onset of lumenogenesis, we purified the cholangiocytes at E16.5 and E18.5. Embryonic livers from SOX9-GFP embryos were dissociated; hematopoietic cells were removed by magnetic cell sorting, and GFP+ cells were FACS-purified. Total RNA was extracted and subjected to RNA sequencing. Expression of apical Aquaporins 1 and 8, Multidrug resistance 1 P-glycoprotein (Abcb1/Mdr1), Na+-independent Cl−/HCO3− exchanger (Slc4a2/Ae2), Cystic fibrosis transmembrane regulator (Cftr) and Na+/HCO3− cotransporter (Slc4a4/Nbc1) was measured. Their expression by cholangiocytes is required for fluid secretion [52, 53] and was detected at E16.5 and E18.5. This indicates that regulators of ion and water transport were present when bile starts to flow (E17.5-E18.5) [7] and even earlier (E16.5) at the onset of lumenogenesis (Fig 4). Hence osmosis is another candidate for lumen formation. In intrahepatic bile ducts, regulated and inducible water influx creates unidirectional advection.

In summary, we identified three candidate mechanisms possibly driving bile duct formation, each of them being compatible with experimental observations: (1) Differential proliferation rates within the cholangiocyte and hepatoblast cell layer within the double layer aligning the portal vein. (2) Apical constriction in at least one of the two cell layers. (3) Secretion of water by osmosis if the salt concentration at the two-layer interface is locally elevated, and fluid leakage is restricted by tight junctions. However, the double layer is constrained on the side of the cholangiocyte layer by mesenchyme enclosing the lumen of the portal vein and on

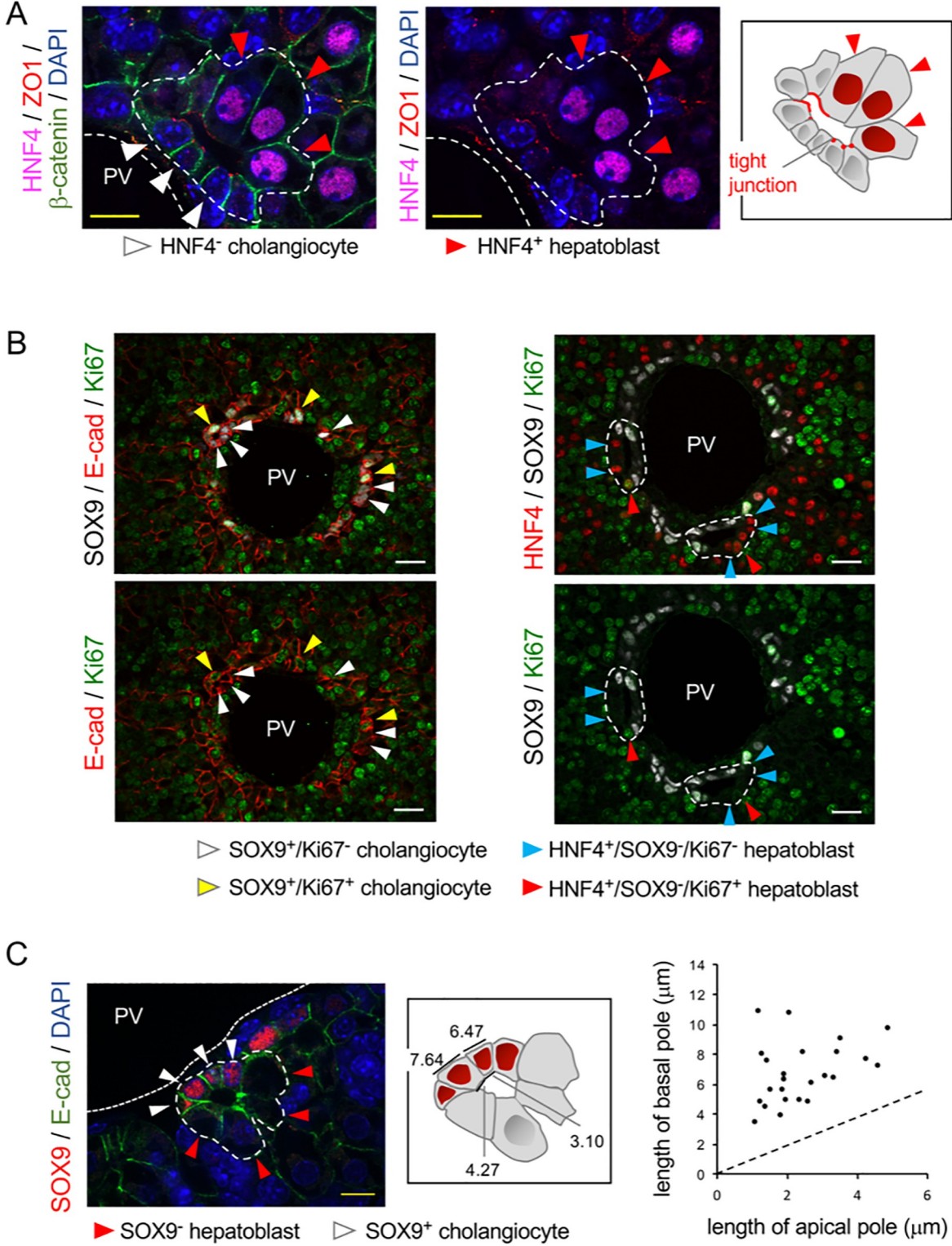

**Fig 3. Morphological features of cells delineating developing biliary lumen.** (A) Tight junctions are detected between HNF4-cholangiocytes and between cholangiocytes and HNF4+ hepatoblasts; but not between adjacent hepatoblasts. (B) Proliferating SOX9+ cholangiocytes and HNF4+ hepatoblasts are detected by Ki67-staining in E16.5 embryonic livers. E-cad: E-cadherin. (C) Apical constriction in cholangiocytes. Developing ducts are delineated by white dotted lines. White size bar, 20 micrometer; yellow size bar 10 micrometer. PV: portal vein.

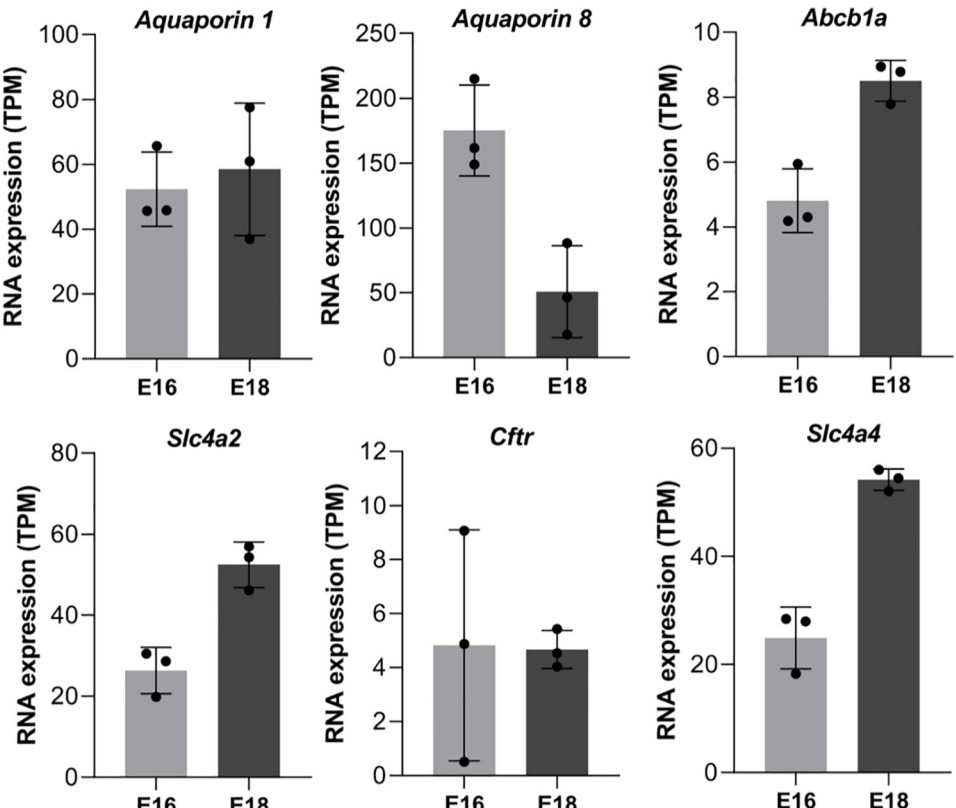

**Fig 4. Expression of ion transporters and water channels during biliary lumenogenesis.** Expression of ion transporters and aquaporins was measured by RNA sequencing of total RNA extracted from purified developing cholangiocytes. Bars are standard deviation (SD).

the side of the hepatoblast layer by a largely unstructured mass of proliferating hepatoblasts, blood vessels and hematopoietic cells that may exert mechanical compressive stress on the double layer and hence counteract lumen formation. The balance of arising forces, proliferating forces, bending forces, deformation and contraction forces, is impossible to estimate by reasoning alone. Hence we implemented a representative tissue section within a computational ABM and simulated each of the three hypothesized mechanisms to explore whether any of the three hypotheses would have to be excluded based on physical interactions. This is possible as the computational model is parameterized by measurable parameters, for which the ranges are largely known.

In a next step we tested whether each of the three candidate mechanisms is able to generate a lumen in an idealized layer before embedding the bi-layer into its natural environment during bile duct formation.

## Computational model: Each mechanism is able to generate a lumen in an isolated double layer

To illustrate the effect of differential proliferation, apical constriction and osmotic forces, we first considered a minimal, idealized subsystem of two layers of each 10 cells which adhere to each other (Fig 5). The model assumes translational symmetry in the direction perpendicular to the represented cut section, as this is the case over several cell diameters in the experiments

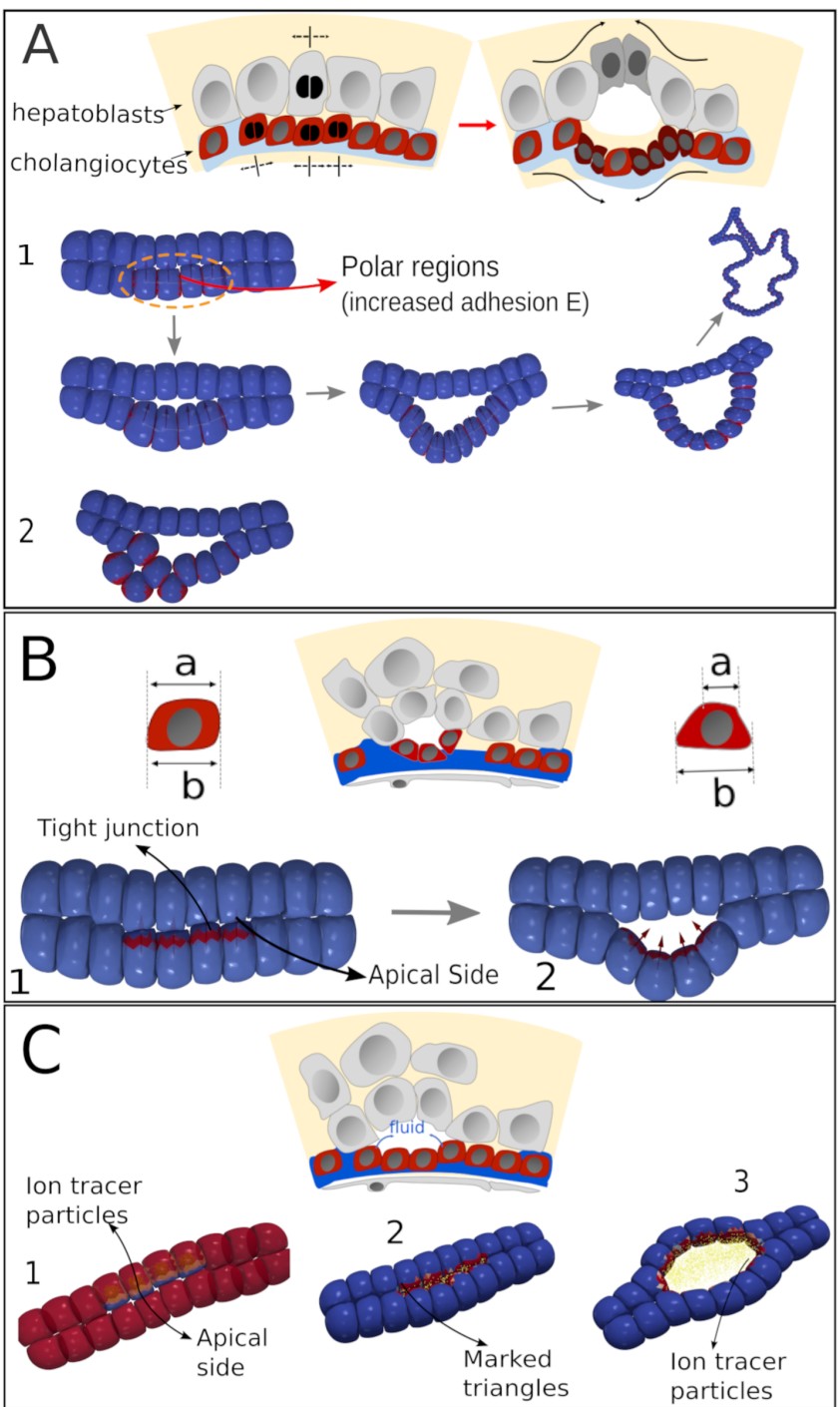

**Fig 5. Possible mechanisms of lumen formation in a cell bi-layer.** (A) Cell division. 1: double layer of cells with 4 dividing cells (indicated by orange dashed line). Cell division goes along PCP. 2: random directed cell division. (B) Apical constriction on the same 4 cells. Note the wedge-like shape of the four center cells with apical membrane marked in red in configuration 2 compared to configuration 1. (C) Osmosis initiated by 4 cells inducing hydrostatic pressure in extracellular space.

in Fig 1A. The system is assumed to be embedded in a fluid medium without external resistance from other cells or ECM. Simulations with this system can help us to understand the effects of these mechanisms in the absence of effects from surrounding cells.

**Mechanism I: Creation of a lumen by pure cell division.** Cell proliferation can introduce a buckling instability in epithelial sheets and formation of cavities (see e.g. [41, 54]). We illustrate this with a simulation of our reduced system. Four cells in the upper layer are selected as proliferating. All other cells remain quiescent. Furthermore, only these 4 cells are assumed to be polarized. The polarization direction is here uniquely determined by the mutual adhesive contact regions between the cells (identified by triangles of the cell surface) between a cell-cell contact of the same type (red marks in Fig 5A) [55]. Thus, for each of these cells the polarization vector is initially parallel to the layer, but can change and is updated along with the relative positions of the cells. In addition, an apical side and basal side are introduced (determined by an apical vector, which is perpendicular to the polar vector). The apical side is oriented towards the other cell layer, while the basal side is oriented outwards. Two cases are now distinguished: (i) When cell division occurs, the division direction is always along the polar vector (Fig 5A). Moreover, the polar contact area is assumed to be populated with a higher intercellular density of molecular complexes forming tight junctions and thus higher in adhesion energy, compared to the average cell adhesion energy. In contrast, the apical side area has almost no cadherins. (ii) The cells are still assumed to be polarized, having still an inhomogenous distribution of adhesion contacts, but this polarization has no effect on the orientation of proliferation, which is here chosen from a random uniform distribution (Fig 5A).

For both cases we assume that only the lower cells can proliferate without any restriction (division time 24h). For case (i) the formation of a convexly shaped lumen after a few cell divisions occurs (Fig 5A and S1 Video). The convex shape of the lumen is conserved even after several cell divisions but eventually, new buckling effects will arise. In contrast, case (ii) does not result in a clear single lumen formation, as cells basically move in random directions due to the proliferation, thereby filling up any previously formed cavity (see Fig 5A and S2 Video). This shows the importance of both cell polarity and polar division in lumen creation, consistent with the findings in ref. [56].

**Mechanism II: Creation of a lumen by apical constriction.** During apical constriction, the surface area of the apical side shrinks due to a local contractile effect of the cortical cytoskeleton. When this effect takes place on several adhering cells simultaneously, a new mechanical equilibrium emerges, driving the layer to bend outwards. Apical constriction has been modeled using vertex models showing that it can significantly contribute to tissue distortion [57–59], and by center-based models to explain gastrulation in sea urchin [41]. Here, we further assume that along with apical constriction, Tight Junctions (TJ) are also present. TJ involve adherent junctions which are usually localized at the apico-lateral borders of epithelial cells [60].

In the minimal system, first the apical vectors of the four upper cells are defined. These point towards the underlying cell layer (red arrows in Fig 5B). The apical vectors define the cone of the apical region in those cells (Fig 5B, dark blue regions). These regions have by default a much smaller adhesion energy ($W_{ap}$) than the rest of the cell surface. On the contrary, the tight junction regions (red triangles in Fig 5B) have a higher adhesion energy. From the moment at which a strong constriction is applied in the apical region a small cavity is forming. No cell division is involved. We note here that when packed in a sheet, such as occurs with the four central cells, the DCM simulations show slightly wedge-like cell shapes (see Fig 5B and S3 Video) as often depicted in textbooks. Altogether, these simulations show that apical constriction of cells can indeed create a small initial cavity.

**Mechanism III: Creation of a lumen by osmotic effects induced by ions.** As previously explained, regulators of ion and water transport could be identified on the apical side of cholangiocytes at the onset of lumenogenesis. This machinery enables to generate differences in the mole fraction of solutes between a cell and extracellular cavities, which could for example have been formed by vesicle exocytosis. This could result in osmotic pressure-driven water flow, until the hydrostatic and osmotic pressures in the cavity and outside are balanced [61, 62]. The excretion of ions and their diffusion is mimicked in the computational model by tracer particles (TP). In the simulation we assume that the four cells are now able to secrete ions (TPs) through a specific region of the cell area (Fig 5C). In agreement with the experiments the "permeable" region is the apical side of the cells, which in turn is determined by the cell's APB vector. As in the previous case, this vector points towards the other cell layer. Right after the simulation has started the tracer particles are created in the cell center and diffuse inside the cell. They gradually move towards the cell boundary and can, mediated by ion channels and transporters, cross the cell membrane on the apical side where there is free extracellular space, contributing to an increased osmotic pressure. (Fig 5C, right). Generally, the osmotic pressure difference -for simplicity assumed here to be constant- triggers water inflow into the initially present small extracellular cavities. The inflowing water exerts a hydrostatic pressure on the cell walls delineating the cavity (marked by red triangles in Fig 5C, middle) and pushes them away, hence increasing the cavity volume. Whether cells from the one layer will be separated from those of the other layer, will depend on the competition between the pressure forces and the surface adhesion forces between the cells. In Fig 5C, right the simulation case is shown where the cell-cell adhesion energy between the two cell layers at the apical side is relatively small compared to the osmotic energy. As a consequence, the cell layer separates from the other layer and a cavity is formed that keeps on growing (see also S4 Video). When the osmotic pressure is finally equilibrated with the tensile adhesive stress between the cells, the lumen stops growing. However, if the osmotic forces were too large, the lumen would eventually burst. Here the presence of tight junctions between the cells helps prevent bursting as they reinforce the cell-cell adhesion and provide a larger mechanical resistance to osmotic forces. Moreover, the tight junctions prevent ions from moving through the cell-cell boundary regions thereby hindering leakage out of the formed cavity. The latter phenomenon can only be modeled approximately at the chosen DCM resolution but by adopting a higher resolution, this would become possible, see section Discussion and conclusions.

## Model: Simulation of lumen formation in bile duct system

In the previous section it has been shown by computational modeling that initial lumen formation can independently be induced by each of the three different hypothesized mechanisms in an idealized system, with no external constraints, to a "minimal" configuration of cells. In the next step we examined whether these mechanisms are able to cause lumen formation if surrounded by tissue, which is the situation during bile duct development in the developing embryo. In addition, since hepatoblasts lining the emerging lumen are gradually replaced by cholangiocytes, the combined effects of "non-mechanical" cues, such as the experimentally observed cell cycle progression and cell signalling, have to be taken into account. Our modeling strategy is to identify the minimal set of mechanisms that are able to cause lumen formation, starting from the time point where no lumen is present (few or no cholangiocytes) until a full lumen completely surrounded by cholangiocytes has been established (see Fig 1).

**Configuration of the in-silico embryonic liver system.** Bile duct formation is guided by a complex interplay of cell signaling and cell mechanics, resulting in initial lumen formation, the formation of tubular branches, and possibly merging of those branches [3]. Simulating this

entire development would require a full 3D simulation demanding large computational power and additional three-dimensional experimental data. Here the focus is on the initial stage of formation of a single bile duct. Because of the translational symmetry along the portal vein, this can be mimicked by simulation of the tissue organisation dynamics in a 2D section of the portal vein area chosen such that the forming lumen lies in the simulated section. In the simulations this is obtained by constraining the cell motion within a plane created by a transversal cut of the portal vein, noticing that the bile ducts develop parallel to the portal vein. This chosen model configuration is representative of the configuration in the experimental images.

To cope with the higher computation time of the DCM, a hybrid model of DCM and CBM has been constructed in which the DCM and CBM cells interact in a mechanically consistent way [37]. The cells initially are labeled as either being endothelial cells, hepatoblasts (and hematopoietic cells), or mesenchymal cells. The mesenchyme and the hepatoblasts in the active segment localized closest to the mesenchyme are represented by a DCM, the hepatoblasts further away from the portal vein are represented by a CBM. The latter cells are assumed to not directly participate in bile duct formation. This turned out to be a self-consistent assumption in that during the simulations no re-arrangements of cells occurred that resulted in direct interactions of CBM cells and DCM cells lining the emerging lumina. The portal vein endothelial cells are represented by CBM cells fixed in space. This is justified as blood flow through the portal vein, which is evidenced on histological sections by the presence of red blood cells [51], would have a similar stabilizing effect. The limited amount of ECM present in intercellular space was not explicitly modeled. However, as in ref. [37], ECM-cell friction has been considered at those surface regions of each cell that are not in direct contact with other cells.

In the experimental images the cholangiocytes appear to be significantly smaller than the surrounding hepatoblasts. Their cell surface ratios (extrapolated from the circumference) varied by up to a factor of 2 to 4 (see Fig 6). The size difference of hepatoblasts to cholangiocytes may be attributed to three main causes: (i) cholangiocytes are likely responsible for excreting salts into the luminal space through the apical side, thereby attracting water. This water

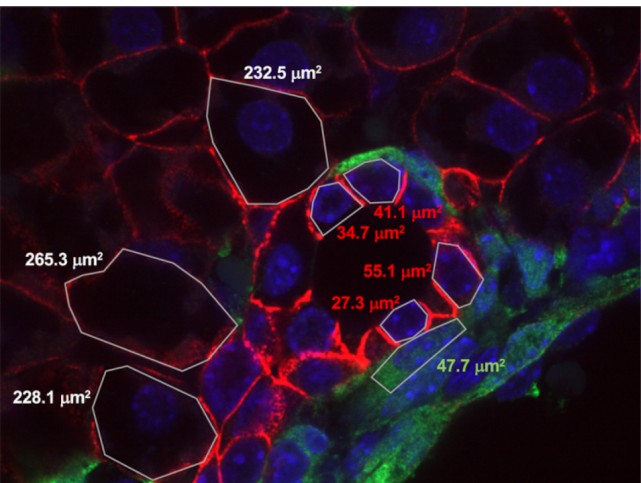

**Fig 6. Detail image of the lumen with indicated cell surface areas.** White numbering: hepatoblasts, red numbering: cholangiocytes, green numbering: mesenchyme. The picture is from an E18.5 mouse liver expressing eYFP in the mesenchyme. Red membrane staining of hepatoblasts and cholangiocytes: E-cadherin; green staining of mesenchyme: eYFP.

content may be partially withdrawn from the cholangiocytes themselves (i-a). On the other hand, aquaporins are also present on the basal side (i-b). The cholangiocytes seem thus to act as a "pump", withdrawing water from regions further away from the bile duct. (ii) Cholangiocytes being neighbors to hepatoblasts may instantly look smaller because they are likely the differentiated daughter cells of a former hepatoblast. Indeed, it was observed that some hepatoblasts adjacent to a cholangiocyte still have a large size but start expressing some cholangiocyte-specific proteins (SOX9+). Upon cell division, they may become cholangiocytes expressing high levels of SOX9+. Hence, we may also assume that cholangiocytes induce active hepatoblast-to- cholangiocyte transformation. (iii) The volume of a mature cholangiocyte is inherently smaller than that of a hepatoblast. This also means that after a hepatoblast divides, the two daughter cholangiocytes have a volume that is relatively large compared to a mature cholangiocyte. As a consequence, the criterion for cell division, i.e. doubling of the initial volume, will be reached relatively quickly and a new cell division will take place shortly after the first cell division. This results in cholangiocytes which are significantly smaller than the original hepatoblasts.

The hypotheses i-b), ii) and iii) have been adopted in our model. However, it is not excluded that hypothesis i-a) is also valid, solely or in combination with i-b). The former could be tested in more depth by adding an extra equation for the change in cell reference volume that describes how much water a cholangiocyte loses during and after its differentiation. However, in the scope of this paper this option has not been considered. Because of hypothesis iii), an initially fast appearance of several new cholangiocytes may occur (Fig 1A). This seems to be in agreement with the experimentally observed configuration (Fig 3A), where four adjacent cholangiocytes in a row are observed, while the other (bigger) cells are still merely hepatoblast.

The cells are initially configured as shown in Fig 1C. The initial size of all the cells is randomly sampled from a Gaussian distribution with average size $d_{mes}$ = 10 $\mu m$ ± 10% for the mesenchymal cells, $d_{hep}$ = 15 $\mu m$ ± 10% for the hepatoblasts. In order to mimic the movement constraints in the tissue model, we have applied a Boundary Conditions (BC) on the outer circumferential side. The tissue is allowed to expand due to cell growth, yet a constant "background" pressure $P_b$ is exerted on to the outer hepatoblasts in radial directions towards the portal vein, mimicking the mechanical resistance of the surrounding tissue (see Fig 1B). The reference value was set to $P_b$ = 50 Pa (personal communication with Adrian Ranga, KULeuven, Belgium), which is of the same order of magnitude as the elastic modulus of liver matrix [63]. On the inner boundary, the fixed endothelial cells (CBM) prevent cells from moving toward the portal vein. More details and technical aspects about the BC are given in the Appendix. An animation of the system can be viewed in S5 Video

The simulations start with the assumption that a lumen originates from a single hepatoblast located in the middle of a segment (Fig 1). At a certain point in time, a differentiation of this hepatoblast into a cholangiocyte is induced, likely mediated by contacts between portal vein-associated mesenchymal cells and adjacent hepatoblasts. The new cholangiocyte undergoes cell cycle progression. The initial size of the cholangiocyte is set to 0.75 times the size of a hepatoblast. The cholangiocyte is assumed to have acquired an apical direction pointing away from the portal vein centre and hence a polar direction perpendicular to the apical direction [40]. After some time, the cholangiocyte divides, creating two daughter cells. These two adherent daughter cholangiocytes are initially proliferating and also become immediately polarized with the same polarity as the mother cell.

We recall that about 20% of the cholangiocytes nearest to the portal vein are proliferating (Ki67+ staining) between E16.5 and E18.5. This is in contrast with the hepatoblasts of which only 11% proliferate. Similar behavior was observed in ref. [64]. As information about the

exact mechanisms maintaining these proliferation rates was lacking, an algorithm was implemented enforcing the experimentally observed rates.

The algorithm keeps track of the number of proliferating cells for each cell type, and randomly picks cells that become quiescent if the observed number of proliferation in the simulation exceeds the experimentally observed number (see S1 Text).

**Simulation of hypotheses.** In this section we present the results of simulations applied to the hypotheses introduced above in the context of a developing embryonic liver (section Computational model: Each mechanism is able to generate a lumen in an isolated double layer). The complexity of the model is increased stepwise, and the simulated lumen shapes and sizes are compared to the experimentally observed ones. All simulations ran for 24 hours after the time point at which the first cholangiocyte appeared. This initial cholangiocyte gets polarized. The basal membrane of the cholangiocyte is oriented towards the mesenchyme (Fig 1A), while the apical vector points away from the PV centre, and the polar vector is aligned with the PV tangent. This can be induced by laminins present in the ECM surrounding the portal vein [65]. The division orientation is thus tangential to the PV, as depicted in (Fig 7A). Notch-like signaling from cholangiocytes to adjacent hepatoblasts is possible, meaning that a hepatoblast can differentiate into a cholangiocyte provided that hepatoblasts are in contact with each other, and that the hepatoblast contacts a free extracellular space (here the duct lumen). The cholangiocyte will try to orient its apical vector to the free extracellular space. The signaling is controlled by a single time parameter $T_N$. We assume that $T_N > 0$ as otherwise the hepatoblasts transform immediately to cholangiocytes, which would be in contrast with the gradual expression of SOX9 observed in Fig 3C (left panel). Here, we postulate that $T_N = 2h$. However, the effect of $T_N$ will be further studied below. For each model and parameter set, 5 simulations were run, each representing a different realization of a stochastic process. The average of these runs were used as comparison.

The models discussed hereafter add one or several components to the reference scenario defined as "Model 0" (see Table 3), where the initial daughter cholangiocytes proliferate further in random directions. In model 0, apical constriction and osmotic effects are absent. The results of the simulation with model 0 show a growth in which few hepatoblasts develop to cholangiocytes, without any cavity formation and lumen growth (snapshot in Fig 7B).

**Model 1.** In model 1 the cholangiocytes become polarized and the division is oriented along the PCP vector (Fig 2E). Both cholangiocyte daughter cells inherit the polarization direction of the mother cell right after cell division. The results with model 1 show the creation of a small cavity, but without clear and stable lumen growth over time (Fig 7C and 7D) as the cavity surface remains smaller than the cross sectional surface of one cell. This is in contrast to the findings of mechanism I (section Model: Testing of three basic mechanisms). The difference can be attributed to the fact that in model 1 the surrounding cells are exerting a pressure preventing further growth of the cavity.

**Model 2.** We extend model 1 with the capacity of the cholangiocytes to perform constriction of the apical side. This involves a ring-like zone of strong constriction with the tight junctions, and a weaker constriction of the apical domain (see S1 Text for details). All other mechanisms are the same as in model 1. Compared to the runs of Model 1, we find an initial stronger signal for cavity area, due to the retraction of the apical domain caused by the mechanical forces of apical constriction. However, this signal is not maintained as it weakens after $t = 12h$ and becomes similar to Model 1 after 24h, see Fig 7C. The reason is that despite the shape formation, the increasing compression due to cell division in combination with the pressure exerted by the surrounding tissue does not allow formation of a stable cavity.

**Model 3.** Model 3 adopts all the characteristics of model 2 and extends the capacity of the cholangiocytes to excrete molecules in the extracellular space inducing an osmotic activity (Fig

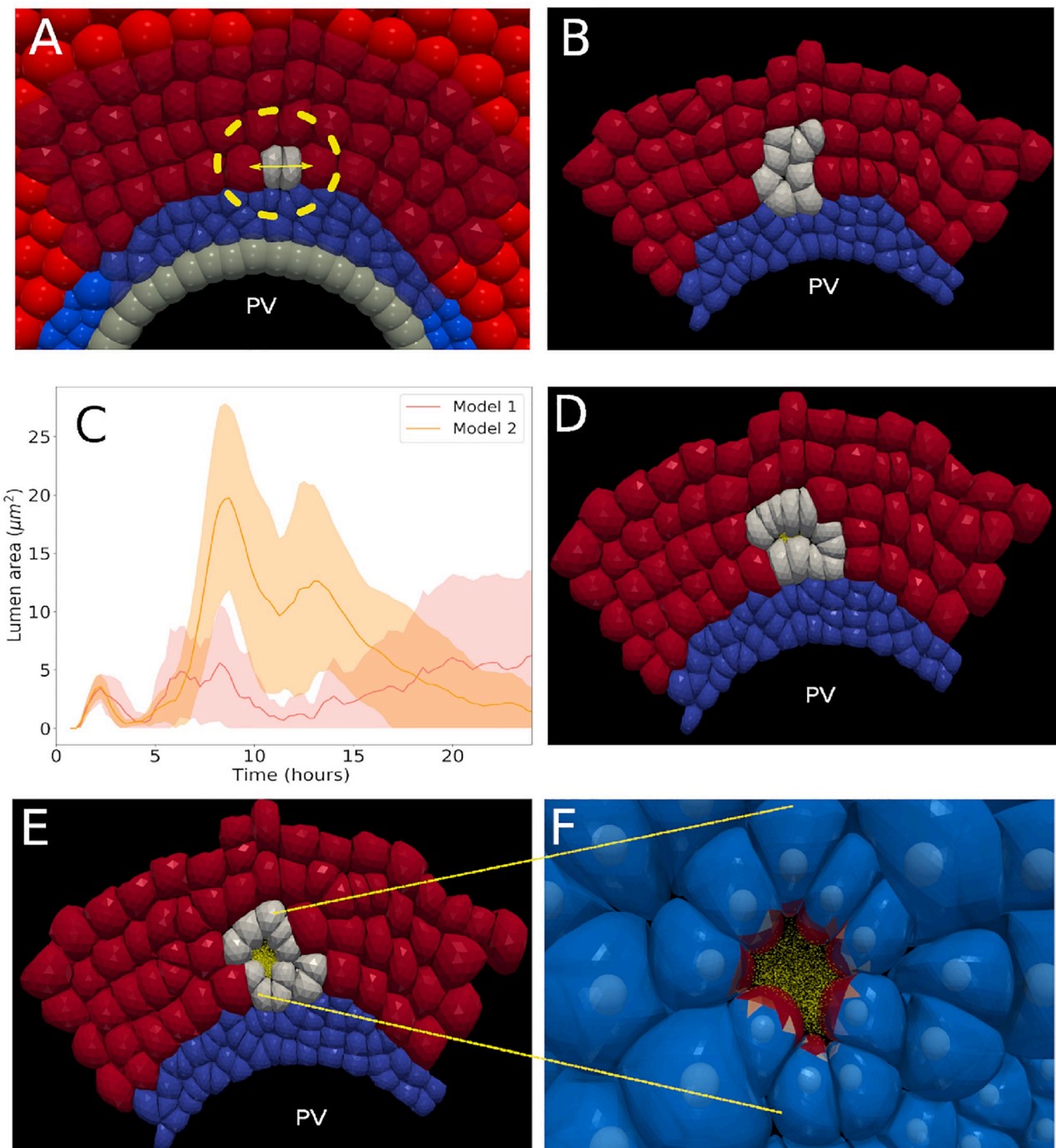

**Fig 7. Simulation results for Models 1 and 2.** (A) Immediately after the initial state, where a cholangiocyte divides in two daughter cells. The division direction is indicated by the arrow. (B) Snapshot of a simulation for Model 0 ($t = 12$). (C) Lumen area versus time for models 1 and 2, for different runs (no osmotic effects present). The solid lines are the average of 5 stochastic realizations of the same parameter set, while the shadowed regions indicate the two times standard deviation interval of these realizations. (D) Snapshot of a simulation for Model 1 ($t = 12$). (E) Snapshot of a simulation for Model 2 ($t = 12$). The red, grey and blue cells indicate hepatocytes, cholangiocytes and the mesenchyme respectively. (F) simulation snapshots for model 2 showing TP's (yellow) and apical sites of the cholangiocytes (red).

**Table 3. Models and the feature they represent.**

| Model | Cell division dir. | apical constriction | Osmotic effects |
|---|---|---|---|
| Model 0 | random | no | no |
| Model 1 | oriented | no | no |
| Model 2 | oriented | yes | no |
| Model 3 | oriented | yes | yes |

8). Expression of ion and water transporters was measured to study possible evidence in favor of the hypothesis that osmotic pressure could be generated at this stage. However, as we have no exact information about the osmotic pressure nor about the osmotic activity over time, we simulate different scenarios with varying pressure, assuming that this pressure remains approximately constant over the considered time period. As shown in Fig 8A and 8C, this hypothesis has a significant impact on the results. For a osmotic pressure $P_L = 25Pa$, lower than the tissue background pressure $P_b = 50Pa$, the lumen size does not become significantly

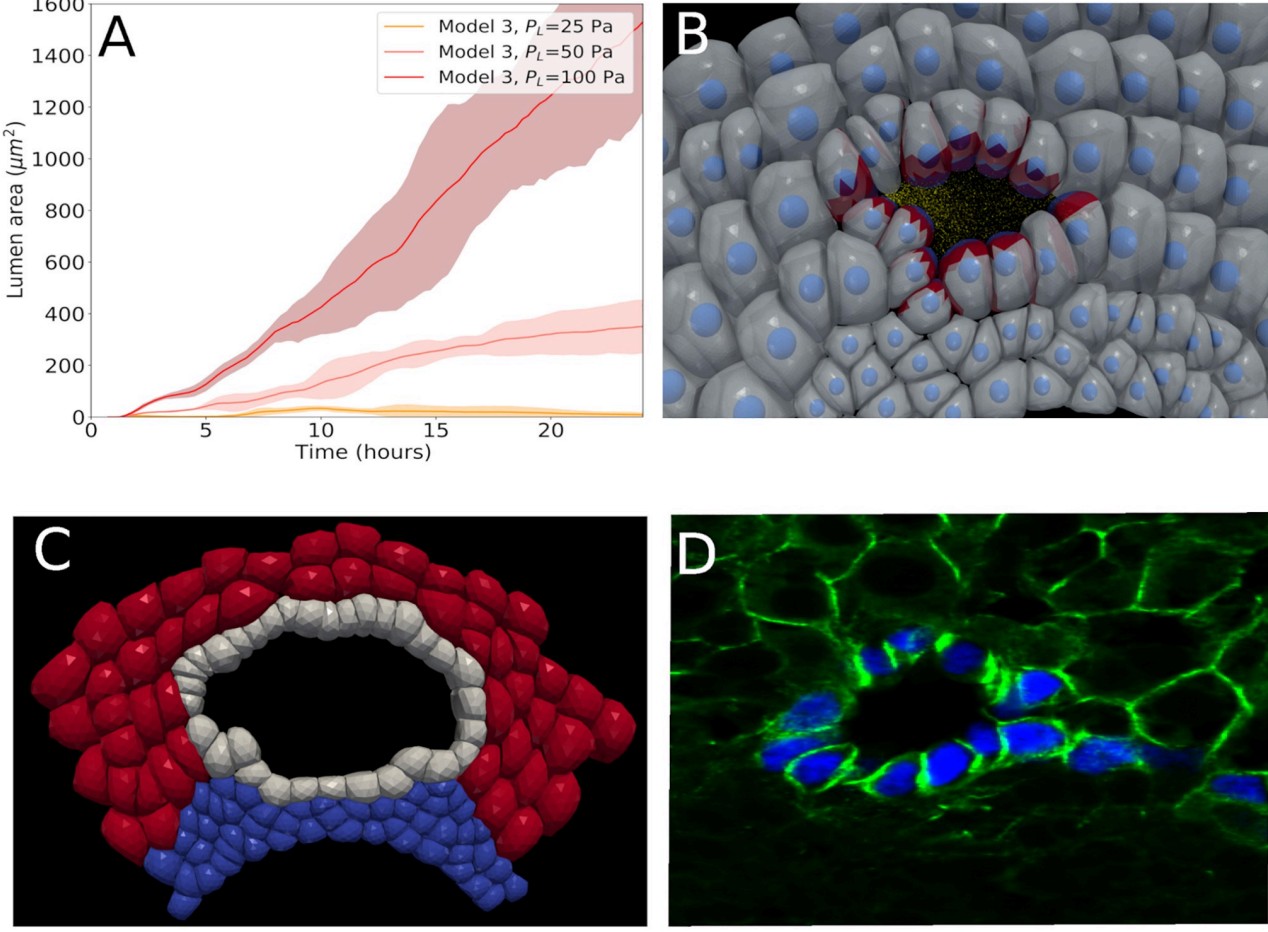

**Fig 8. Simulation results for Model 3.** (A) lumen area versus time for different osmotic pressures. The solid lines are the average of 5 stochastic realizations of the same parameter set, while the shadowed regions indicate the minimum and maximum values of these realizations. (B-C) Snapshot of a simulation for Model 3 ($P_L = 50Pa$ and $P_L = 100Pa$ respectively). In B, the red zones indicate increased cell-cell adhesion due to presence of TJ. (D) Picture of typical bile duct lumen for an embryo at E18.5 (SOX9, blue; beta-catenin, green).

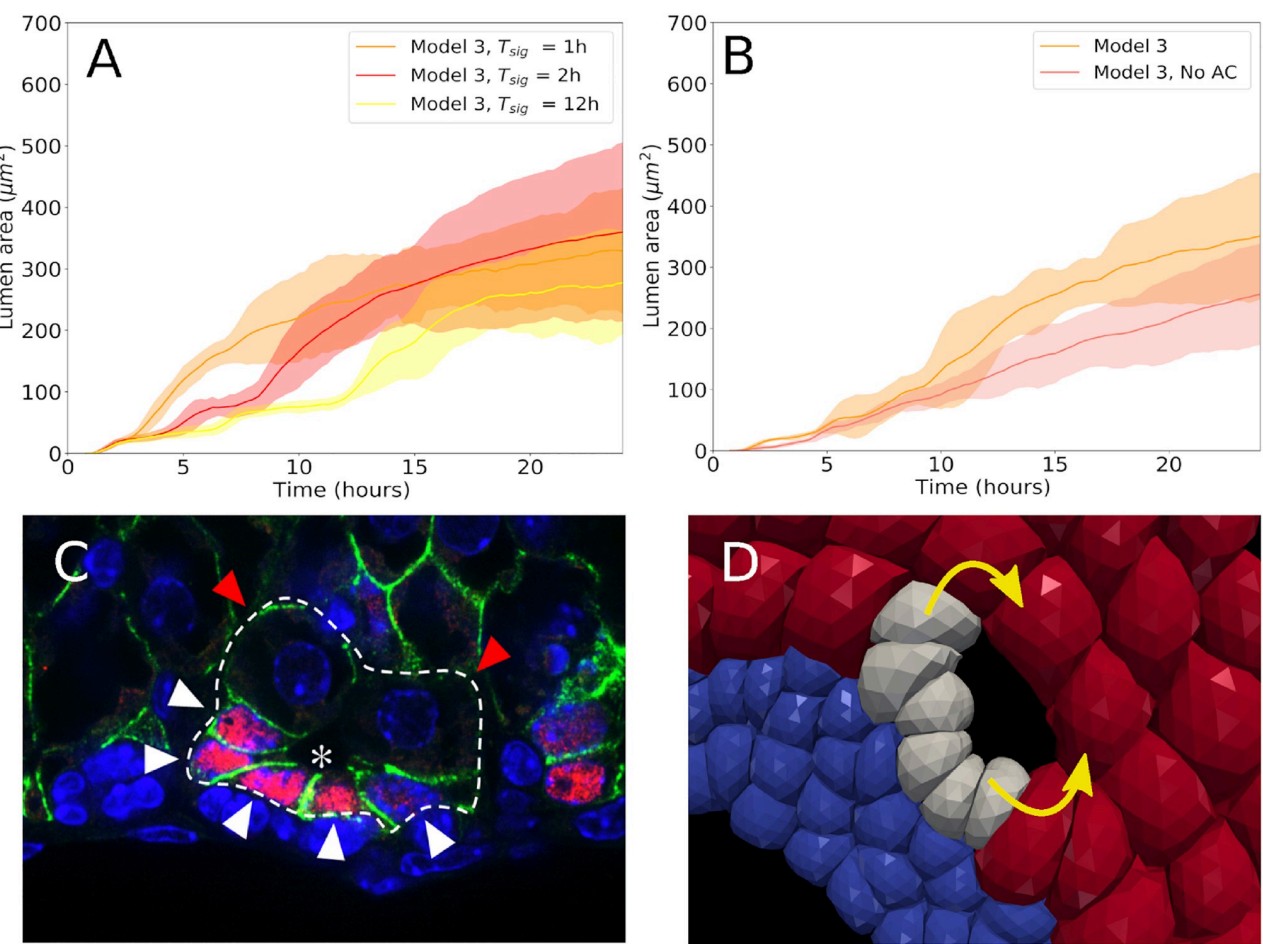

**Fig 9. Simulation results for Model 3, assuming $P_l = 50Pa$: Lumen area versus time.** (A) influence of cell-to-cell signalling times $T_{sig}$. (B) influence of absence or presence of Apical constriction (AC). The solid lines are the average of 5 stochastic realizations of the same parameter set, while the shadowed regions indicate the two times standard deviation interval of these realizations. An increase in lumen area from zero to $\sim 350\mu m^2$ in 24 hours is realistic since the area of lumina near the hilum ranged from 7 to 36 $\mu m^2$ at E16.5 and from 116 to 674 $\mu m^2$ at E18.5 (n = 10 at each stage) (C) Microscopic picture of small lumen with cholangiocytes (white arrowheads) and hepatoblasts (red arrowheads). Tissue section is stained to detect SOX9 (red), beta-catenin (green) and nuclei (DAPI, blue). (D) Snapshot of simulation showing a configuration comparable to that in panel C. The yellow arrows indicate the direction of cell-cell signalling during the formation.

larger than the lumina obtained with Model 1 and Model 2, while similar as in those cases it tends to collapse at the end of the period. However assuming $P_L = 50Pa$ a drastic increase in lumen size can be observed. Interestingly, the lumen size becomes almost stable for $P_L = 50Pa$ while it continues to increase sharply for $P_L = 100Pa$. This shows that a control of osmosis could be an important factor in the lumen stability. In this regard it is also interesting to see how much of the lumen formation can be attributed to apical constriction. To study this question, simulations of Model 3 ($P_L = 50Pa$) have been performed for which AC has been omitted. The results, shown in Fig 9B, indicate that the presence of AC (as compared to no AC) speeds up the lumen formation, although it does not seem to play a major role to finally establish a stable lumen. In conclusion, oriented cell division in combination of osmotic control is able to generate a stable lumen. (An example animation of a simulation can be viewed in S6 Video).

Using Model 3 we also studied how lumen formation is influenced by the cell-to-cell signalling time, i.e the average time it takes for a cholangiocyte to induce differentiation of a

neighboring hepatoblast (Fig 9C and 9D). We recall that only hepatoblasts delimiting an initial (possibly initially very small) lumen can differentiate. The signalling time $T_{sig}$ was varied between $1h$ and $12h$. Here, the model runs show (Fig 9A) that the influence is limited to the initial stage of lumen formation, where a short $T_{sig}$ (= 1h) favors a quicker onset of lumen formation as compared to the case $T_{sig}$ = 12h. This can be mainly attributed the earlier cell division of the cells bordering on the lumen, cfr. point ii) in section Configuration of the in-silico embryonic liver system. Hence, the cell-to-cell signaling time seems to determine the onset of cell division and thereby the time at which the lumen becomes visible. For larger simulation times the difference between the model runs becomes much smaller, as at this point all the cells delimiting the lumen have become cholangiocytes and there is no further differentiation.

There are also cell specific mechanical parameters which will potentially deviate the force balance in this bile-duct system, and thus potentially the lumen dynamics. The most obvious one here is cell to cell adhesion energy. Secondly, cell stiffness may influence the mechanical balance as well. We have performed additional simulations here to illustrate the effect of those parameters. First, we have performed a simulation set in which we assume a higher global adhesion energy between the cells, but whereby the cadherin density on the apical poles of the cells remains low. An overall higher adhesion energy between the cells modifies the forces needed to separate cells and thus the origin of the lumen and evolution its size. The effect of a two times higher global adhesion, on the lumen size is shown below in (Fig 10A). As could be expected here, a higher adhesion energy decreases the lumen size compared to the nominal adhesion energy value, a consequence of the higher forces between the cells that need to be overcome by the osmotic pressure. We here also verified the case where local adhesion would be higher than expected, i.e. at the apical sides of the cells. In particular, we asked ourselves whether the observed low density of cadherins on the apical side compared to the cell to cell junctions are key to a lumen initiation. To this end, we ran simulations in which the apical specific adhesion energy was increased from $W_{ap} = 10^{-6}\ J/m^2$) to the global nominal adhesion energy ($W_{ap} = 9.10^{-4}\ J/m^2$) between the cells. This mimics the scenario in which one would observe a homogeneous cadherin distribution. The result shown in (Fig 10B–10C) now shows an overall significant drop of the lumen size. However, the results are lumen pressure dependent. If the nominal value of the lumen pressure are assumed, no lumen develops. If a higher lumen pressure is assumed $P_l$ = 70$Pa$, several realizations suggest that a small lumen may develop (Fig 10C). This suggests a critical importance of the balance between apical adhesion energy ans osmotic pressure: if the adhesion forces between the apical sides of the cells would be too high, lumen initiation may be impeded, hence preventing normal bile duct development.

Lastly, we performed simulations in which the cells have different mechanical properties. We looked here at the case in which the cells are of a stiffer type. For this we set double values of the cortical stiffness and the volume control stiffness, compared to the nominal values. The results did not show a significant effect on lumen size. Absence of such effect could be expected, as the internal forces inside the cell are changed here, but not the ones between the cells.

## Discussion and conclusions

In the presented work, a high resolution model was established to better understand which mechanisms are capable of forming a lumen during bile duct development, using a combined data-based and computational model-based strategy. The selection of hypotheses in the computational model was informed by investigations at the molecular level that suggest several mechanisms controlling lumen formation, namely apico-basal polarization, apical

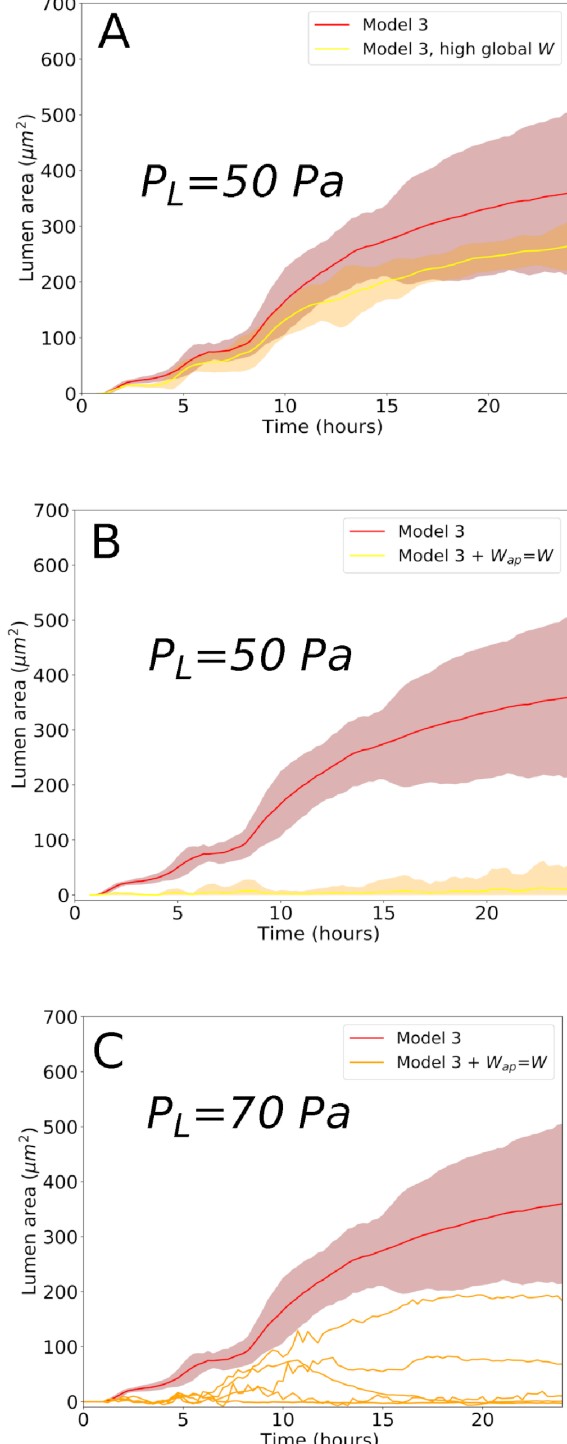

**Fig 10. Simulation results compared to Model 3 with nominal parameters.** (A) Effect of a higher global cell-cell adhesion energy value. (B-C) Effect of an apical adhesion energy value equal to the global cell-cell adhesion energy ($W_{ap} = W$), for $P_l = 50Pa$ and $P_l = 70Pa$ respectively. In the last figure, only the individual realizations are shown for clarity reasons.

constriction, cell-cell repulsion, creation of apical surfaces, secretion of ions or vesicle exocytosis [10]. In a first step, morphometric data for the building of the computational model of biliary lumenogenesis were collected with a focus on quantifying cell proliferation and apical constriction. Expression of ion and water transporters was measured to identify possible evidence for the hypothesis that an osmotic pressure could be generated at the earliest stage of biliary lumenogenesis. Further, bile duct lumen formation occurs as a variant of cord hollowing. Standard cord hollowing is characterized by the creation of a lumen within a cylindrical cord of cells [9]. In bile ducts, the lumen forms between cells that have distinct phenotypes, namely between cholangiocytes and hepatoblasts [5]. Since hepatoblasts differentiate to cholangiocytes during biliary lumenogenesis, the dynamic phenotypic changes of cells lining the developing lumen was equally considered.

In the second step the computational model was established to unravel and quantify mechanisms that support lumenogenesis. This consists in a single-cell (agent)-based approach, in which every cell is represented in spatial detail, taking into account the physical forces that determine the cell's shape and motion. The forces that play an essential role in lumen formation during embryonic development allegedly are those that originate from (1) pure cell division, (2) apical constriction, or (3) osmotic effects. The goal was to quantify the extent to which each of the mechanisms may influence the lumen formation.

To verify the effects of individual physical forces originating from cell division, apical constriction or osmosis, we have first built an isolated, minimal system of adhering cells freely floating in a liquid medium. Our simulations in such an idealized system have shown that each of the three mechanisms can indeed induce lumen formation, provided that the cells are polarized and each of the mechanisms is properly oriented, which in the model was implemented as a polarization vector providing a direction. However, in a real embryonic tissue, cells feel continuous "background forces" from the other growing cells. To take these forces into account an in-silico model mimicking the tissue micro-architecture around the portal vein has been built, in which the various cell types, namely mesenchyme, cholangiocytes and hepatoblasts, have been included. The underlying assumption was that the bile duct originates from a single hepatoblast that differentiates into a cholangiocyte, thereby acquiring the capacity to signal to neighboring hepatoblasts. Our model parameters were informed by the experimentally observed proliferation rates of the hepatoblasts and cholangiocytes. Similar to the minimal system, three submodels have been proposed that implement the individual effects of the three mechanisms. From the sampling of 24h simulation runs the conclusions were: (i) directed cell division alone can initiate a cavity but cannot maintain it during 24h; (ii) including apical constriction in the cholangiocytes improves initial cavity formation but did not give a stable lumen growth. In both cases the background pressure forces are too high and induce a collapse of the cavity; (iii) directed cell division combined with apical constriction and induced osmotic effects of the cholangiocytes creates a stable lumen provided that the osmotic pressure has approximately the same magnitude as the background pressure of tissue. A too low osmotic pressure resulted in a collapse of the cavity whereas a too high one resulted in an unrealistically large growth of the lumen. We hypothesize here that the cholangiocytes excrete ions that serve as signaling molecules; (iv) the cell-to-cell signaling period (the time between sending the signaling and responding to it) controls the time needed for a cholangiocyte to induce differentiation of an adherent hepatoblast. This may play a role in the rate of lumen formation, but does not affect the lumen's final size. In a former ordinary differential equation-based model, Gin and co-workers found a control of cyst lumen size in vitro when elastic tensile stress balanced osmotic pressure whereby cell proliferation occurs in response to lumen expansion [61]. This mechanism is largely in line with our findings. Lumen formation in our work is studied within

the framework of a single-cell model which permits to include cell-cell adhesion forces and cell-to-cell signalling.

Cell signaling time, which impacts temporal control of differentiation of hepatoblasts to cholangiocytes, also emerged from the modeling as a key regulator of lumen formation. This is not surprising since differentiation consists in acquisition of cell-type specific form and function, and these include the above-mentioned polarization and secretory capacity of the cholangiocytes. The overarching role of differentiation highlighted in our modeling is supported by the analyses of congenital malformations of the bile ducts in humans. Congenital ductopenia, or paucity of the bile ducts, is indeed observed in rare human diseases, Alagille syndrome being the best studied among them. It is characterized by the absence of bile duct formation and results from aberrant Notch signaling consecutive to mutations affecting the *JAGGED1* or *NOTCH2* gene [66]. Mouse models knockout for Notch signalling effectors enabled to identify a critical lumenogenic role of hepatoblast to cholangiocyte differentiation [12]. Such mouse models also underline the importance of Notch-mediated control of cholangiocyte polarity in shaping the architecture of the epithelium lining the ducts and determining lumen size and maintenance [67]. Since Notch signaling stimulates expression of genes normally located at the apical pole of the cholangiocytes, including the chloride transporter CFTR, it is likely that deficient lumen formation associated with perturbed Notch signaling results in part from perturbed osmosis. Moreover, Notch signaling functions also by stimulating expression of the transcription factor *HNF1b*, a transcription factor known to control polarity genes [12, 68], thereby establishing a molecular cascade between Notch, differentiation and polarity. Mice knockout for *Hnf1b* display aberrantly-shaped cholangiocytes and enlarged and irregular lumen at the onset of bile duct formation, but these lumina eventually collapse leading to absence of well-defined bile ducts, as in patients with deficient *HNF1b* gene [69, 70]. Whether this relates to abnormal osmosis was not determined, but again is very likely. Together, these observations support the importance of our modeling to test hypotheses explaining how bile ducts may fail to form in human disease.

Our DCM is built in a modular fashion such that cellular detail can be added or removed easily. One could, for example, add components representing an internal cytoskeleton, or add more degrees of freedom to the structure, to obtain a more accurate representation of cell shape. Although these will inevitably bring more complexity to the model, they may become necessary when studying problems such as bile canaliculi formation which demand a finer scale.

## Supporting information

**S1 Text. This text contains more detailed information about the model algorithms and parameters.**
(PDF)

**S1 Video. Animation of simulation.** Cell division provoking cavity formation: Hypothesis I, case polar directed division.
(MP4)

**S2 Video. Animation of simulation.** Cell division provoking cavity formation: Hypothesis I, random directed division.
(MP4)

**S3 Video. Animation of simulation.** Apical constriction provoking cavity formation: Hypothesis II.
(MP4)

**S4 Video. Animation of simulation.** Osmotic effects provoking cavity formation: Hypothesis III.
(MP4)

**S5 Video. Animation of simulation.** Whole system simulation with bile duct formation (Model 3).
(MP4)

**S6 Video. Animation of simulation.** Close-up of simulation with bile duct formation (Model 3).
(MP4)

**S1 Data. Data points generated by simulations.** All figures with simulation data can be generated using the python script delivered for each figure.
(ZIP)

## Acknowledgments

We would like to thank Adrian Ranga for the useful discussion about liver tissue elasticity, and Nicolas Dauguet for help with cell sorting. The software TiSim generated by the group of D. Drasdo was extended to perform the simulations in this paper.

## Author Contributions

**Conceptualization:** Paul Van Liedekerke, Frédéric P. Lemaigre, Dirk Drasdo.

**Formal analysis:** Paul Van Liedekerke, Lila Gannoun, Axelle Loriot, Frédéric P. Lemaigre, Dirk Drasdo.

**Funding acquisition:** Frédéric P. Lemaigre, Dirk Drasdo.

**Investigation:** Paul Van Liedekerke, Lila Gannoun, Dirk Drasdo.

**Methodology:** Paul Van Liedekerke, Lila Gannoun, Axelle Loriot, Dirk Drasdo.

**Project administration:** Frédéric P. Lemaigre, Dirk Drasdo.

**Software:** Paul Van Liedekerke, Tim Johann.

**Supervision:** Frédéric P. Lemaigre, Dirk Drasdo.

**Validation:** Paul Van Liedekerke, Dirk Drasdo.

**Visualization:** Paul Van Liedekerke, Lila Gannoun.

**Writing – original draft:** Paul Van Liedekerke, Lila Gannoun, Axelle Loriot, Frédéric P. Lemaigre, Dirk Drasdo.

**Writing – review & editing:** Paul Van Liedekerke, Frédéric P. Lemaigre, Dirk Drasdo.

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
