## [Decision Letter · Decision Letter 0]

14 May 2021

Dear Dr Van Liedekerke,

Thank you very much for submitting your manuscript "Quantitative modeling  identifies critical cell mechanics driving bile duct lumen formation" for consideration at PLOS Computational Biology.

As with all papers reviewed by the journal, your manuscript was reviewed by members of the editorial board and by several independent reviewers. In light of the reviews (below this email), we would like to invite the resubmission of a significantly-revised version that takes into account the reviewers' comments.

We cannot make any decision about publication until we have seen the revised manuscript and your response to the reviewers' comments. Your revised manuscript is also likely to be sent to reviewers for further evaluation.

Sincerely,

David Umulis

Associate Editor

PLOS Computational Biology

Jason Haugh

Deputy Editor

PLOS Computational Biology

Reviewer's Responses to Questions

**Comments to the Authors:**

Reviewer #1: see attachment.

Reviewer #2: My expertise is not in biomechanics and I therefore focus my comments on agent-based model implementation and analysis. The authors present an agent-based model of bile duct lumen formation. The model builds on prior work and systematically explores biological hypotheses. The work is well-presented and detailed. However, some clarifications would help the reader evaluate their findings and put them into context.

• One key feature of ABMs is stochastic behavior. Most results (e.g. in the idealized system) all seem to show representative images but it is not clear how consistent these behaviors are? I.e. what is the impact of stochastic behavior over multiple simulations with the same parameter set? Some variation is shown in figs 7 and 8 for the full model but it is not stated what the shaded regions represent.

• The authors give good motivation and citations for parameter estimates in Table 2. However, there is no uncertainty analysis to quantify how parameter uncertainty would impact their conclusions. This would be another contributor to variation in their simulation results.

• For mechanism I in the idealized system – the difference between the two cell layers are the that one proliferates and the other doesn’t at all. When in the data it showed that both layers proliferated but at different rates…does this difference in assumption impact their findings or is it just dependent upon relative proliferation between the two cell layers?

• Hydrostatic pressure is generated on cell surface triangles based on their contact with tracer particles. Does this pressure depend on the time in contact with tracer particles? E.g. could a triangle become ‘marked’ if a tracer particle diffuses over its surface and therefore is only in contact for a short period of time? Or asked a different way…how long does the osmotic pressure last? Do these ‘marks’ from the tracer particles accumulate to account for ion concentrations?

• The discussion and conclusions largely restate the results sections and fail to connect their results to broader literature. E.g. are there any known genetic factors associated with bile duct dysfunction that is involved in any of the processes they describe? Do any of their ‘failed’ lumen formation studies resemble in vivo observations in malformation or disease? What studies would be helpful in the near future to take advantage of their observations? Are there any other developmental processes that are known to operate in similar ways?

• How do the simulated time courses (e.g. of lumen area) compare to in vivo results? Specifically, is an increase in lumen area from zero to ~350um^2 in 24 hours (fig. 9A) realistic?

• For Figure 4 and Hypothesis III: the figures show results for E16 and E18 but only E16 is discussed in the text. The authors just state that the figures show that these transporters are being expressed at E16 but there is no baseline or control experiment with which to compare. E.g. is there a time point prior to bile duct formation where you would expect these transporters to not be active? Without some reference or control these results are difficult to interpret.

• Grammatical errors throughout need to be corrected.

Reviewer #3: This is an impressive interdisciplinary paper focussing on modelling aspects of bile duct networks. Specifically investigating the biophysical mechanisms at work during initial bile duct lumen formation during embryogenesis.

The authors use two coupled individual-based, force-based models - a Center-Based Model (CBM) and a Deformable Cell Model (DCM) in order to do this. This approach allows the modelling of both cell-cell interactions and lumen formation in biliary morphogenesis.

The DCM in this paper further develops and extends their previous sophisticated modelling approach through the inclusion of three novel features: (i) Apical-Basal Polarity in each cell; (ii) the modelling of Tight Junctions (TJ) between cells; (iii) Tracer Particles (TP) mimicking osmotic effects.

The authors investigate three different mechanisms hypothesised to contribute to bile duct lumen formation, looking at the individual effects of each of the proposed mechanisms, namely: (i) coordinated cell division; (ii) apical constriction, and (iii) osmotic effects. The computational simulations show that each of these mechanisms can create a lumen in an idealised system without boundary conditions.

Next, guided by the quantification of morphological features and expression of genes in developing bile ducts, the authors construct an in silico system representing a part of the lobule containing the portal vein and surrounding tissue. Using this architecture, the authors use their individual-based model to simulate the effects of the above three mechanisms, both individually and also combined. The results of the computational simulations show that it is necessary for these mechanisms to be coupled together in order to create an initial lumen and then further lumen growth.

The results of the computational simulations have shed light on the underlying biological system (bile duct lumen formation) in a way that would have been very difficult to determine via experiments alone.

The paper is an excellent example of quantitative, predictive and insightful modelling and I recommend publication.

There are some minor typographical errors which should be corrected:

Abstract: This model permit realistic simulations  This model permits realistic simulations

pg. 4, line 83 hypothetised  hypothesised

pg. 5, line 97: Université catholique de Louvain  Université Catholique de Louvain

**Have the authors made all data and (if applicable) computational code underlying the findings in their manuscript fully available?**

Reviewer #1: **No: **1. Source code is not supplied. The models described in the paper can not be validated or reused by others.

2. In the submission documents the authors state "RNAseq data in Fig.4: They are in a public repository. We state in the text that: The RNA-seq data have been deposited in the Gene Expression Omnibus (GEO) database and assigned the identifier GSE163062. The following link has been created to permit the review of record GSE139938 while ensuring it remains private:

https://www.ncbi.nlm.nih.gov/geo/query/acc.cgi?acc=GSE163062, secure token for reviewer: wrahyagizfafdcp"

I can access the data in GEO but the text described above, with the link to GEO, is not in the manuscript.

Reviewer #2: **No: **It was not clear if the model code or executable was publicly available.

Reviewer #3: Yes

PLOS authors have the option to publish the peer review history of their article (what does this mean?). If published, this will include your full peer review and any attached files.

Reviewer #1: **Yes: **James P Sluka

Reviewer #2: No

Reviewer #3: No
---

## [Decision Letter · Decision Letter 1]

16 Nov 2021

Dear Dr Van Liedekerke,

We are pleased to inform you that your manuscript 'Quantitative modeling  identifies critical cell mechanics driving bile duct lumen formation' has been provisionally accepted for publication in PLOS Computational Biology.

We also note that, according to the Editorial Staff, the code sharing issue raised by Reviewer #1 will need to be addressed in the next steps, prior to publication.

Best regards,

David M. Umulis

Associate Editor

PLOS Computational Biology

Jason Haugh

Deputy Editor

PLOS Computational Biology

Please follow current journal guidelines for code and data availability.

Reviewer's Responses to Questions

**Comments to the Authors:**

Reviewer #1: This reviewer appreciates the effort the authors put in to responding to my previous comments. Most of my concerns, issues and questions have been adequately addressed. The corrections in the text and figures have improved the readability and the impact of the manuscript.

The Video #6 now plays correctly for me.

My only remaining concern is the lack of source code and the lack of a Data Availability Statement in the manuscript.

It appears to me that his manuscript was submitted sometime prior to March 1, 2021. At the end of March 2021 Plos Comp Bio revised their code sharing policies stating that code must be made available of there must be an explanation of why it is not must be included in the manuscript.

https://journals.plos.org/ploscompbiol/s/code-availability Quoting:

“…enhanced code sharing policy for all papers submitted from 30 March 2021.”

“In alignment with our data availability policy, PLOS Computational Biology requires authors to make all author-generated code directly related to their study’s findings publicly available without access restriction at the time of publication unless specific legal or ethical restrictions prohibit public sharing of code. In these cases, authors must indicate how others may request access to the code. Access to code must be described in the Data Availability Statement. Relevant code should be available to editors and reviewers at the time of submission and throughout the editorial process, but does not need to be publicly shared prior to acceptance.”

Since this manuscript does not contain links to source code it is not possible to fully test the model presented in the paper. I have no doubt of the skill of the authors, the lead authors have a long history of creating high quality code to implement complex biological models. Furthermore, I do not know which “priority date” should be applied to a manuscript, the original submissions date, which is before the change in code sharing policy, of the date the revision was submitted, which is after the change in code sharing policy.

The importance of the lack of code sharing I leave to the Journal.

I believe the manuscript does need a Data Availability section.

Reviewer #2: The authors thoroughly responded to all reviewer comments including significant new simulations and updates to the text.

Only one minor comment: in Figure 10C for the modified model results (yellow curves) the authors plot individual simulations, when figures 10A and B show shaded regions. The authors do acknowledge the difference in the caption, but in my view the reason for the difference needs to be included.

**Have the authors made all data and (if applicable) computational code underlying the findings in their manuscript fully available?**

Reviewer #1: **No: **The manuscript does not contain a data availability statement. I believe that must be included prior to acceptance.

My only remaining concern is the lack of source code. It appears to me that his manuscript was submitted sometime prior to March 1, 2021. At the end of March 2021 Plos Comp Bio revised their code sharing policies stating that code must be made available of there must be an explanation of why it is not must be included in the manuscript.

https://journals.plos.org/ploscompbiol/s/code-availability

Since this manuscript does not contain links to source code it is not possible to fully test the model presented in the paper. I have no doubt of the skill of the authors, the lead authors have a long history of creating high quality code to implement complex biological models. Furthermore, I do not know which “priority date” should be applied to a manuscript, the original submissions date, which is before the change in code sharing policy, of the date the revision was submitted, which is after the change in code sharing policy.

If the lack of code sharing is accepted to the Journal then I think the manuscript should be published.

Reviewer #2: Yes

PLOS authors have the option to publish the peer review history of their article (what does this mean?). If published, this will include your full peer review and any attached files.

Reviewer #1: **Yes: **James P Sluka

Reviewer #2: No

---

## [Editor Report · Acceptance letter]

28 Jan 2022

PCOMPBIOL-D-21-00553R1 

Quantitative modeling  identifies critical cell mechanics driving bile duct lumen formation

Dear Dr Van Liedekerke,

I am pleased to inform you that your manuscript has been formally accepted for publication in PLOS Computational Biology. Your manuscript is now with our production department and you will be notified of the publication date in due course.

With kind regards,

Livia Horvath
